# Clipping Bottleneck:
# Stabilizing RLVR via Stochastic Recovery of Near-Boundary Signals

**Shuo Yang** [1 2]  **Jinda Lu** [2]  **Chiyu Ma** [2 3]  **Kexin Huang** [2]  **Haoming Meng** [2 4]  **Qihui Zhang** [1]  **Yuyang Liu** [1]
**Bolin Ding** [5]  **Guoyin Wang** [2]  **Li Yuan** [1]  **Jingren Zhou** [5]

## Abstract

Reinforcement Learning with Verifiable Rewards (RLVR) has emerged as a central paradigm for scaling LLM reasoning, yet its optimization often suffers from training instability and suboptimal convergence. Through a systematic dissection of clipping-based GRPO-style objectives, we identify the rigid clipping decision induced by hard clipping as a key practical bottleneck in the studied RLVR setups. Specifically, our analysis suggests that informative signals can lie in the **near-boundary** region just beyond the clipping threshold, and are therefore discarded by the standard hard-clipping rule. Motivated by this diagnosis, we propose **Near-boundary Stochastic Rescue (NSR)**, a minimal, plug-and-play modification that stochastically retains these slightly out-of-bound tokens to recover lost signals. While NSR, via stochastic sampling, can be interpreted as inducing an implicit gradient decay in expectation, our ablations reveal that its stochastic, boundary-local rescue mechanism is consistently more effective than deterministic gradient decay. Validated by extensive experiments across model sizes from 7B to 30B and both dense and MoE architectures, as a plug-and-play solution, NSR substantially improves training stability and delivers consistent gains over strong baselines such as DAPO and GSPO. Our code is publicly available at https://github.com/qwenpilot/NSR.

## 1. Introduction

Reinforcement Learning with Verifiable Rewards (Guo et al., 2025; Jaech et al., 2024; Meng et al., 2026; Ma et al., 2026) has emerged as the critical engine for scaling the reasoning capabilities of Large Language Models (Yang et al., 2025a; Team et al., 2025; Comanici et al., 2025). By leveraging deterministic verifiers in domains such as mathematics and coding, this paradigm enables models to iteratively refine their reasoning paths through outcome-based supervision. In practice, clipping-based GRPO-style (Shao et al., 2024) objectives based on importance ratios—such as GRPO (Shao et al., 2024) and DAPO (Yu et al., 2025)—are widely adopted due to their reliable implementations and theoretical justification. Despite their popularity, these methods often exhibit significant instability and suboptimal convergence: training is hypersensitive to hyperparameters (Liu et al., 2025b), prone to entropy collapse or spikes (Wu et al., 2025; Cui et al., 2025), suffers from poor reproducibility (Hochlehnert et al., 2025), and frequently oscillates during optimization (Hao et al., 2025).

To mitigate these issues, prior research has primarily focused on refining training dynamics through empirical statistical indicators. Common strategies range from modulating exploration via token-level metrics, such as low-probability (Yang et al., 2025b; Huang et al., 2025; 2026) or high-entropy tokens (Wang et al., 2025b), to differentiating updates based on sample-level attributes like correctness (Zhu et al., 2025) and difficulty (Ji et al., 2025). While effective in practice, these approaches mostly act as **external adjustments** to the training signals, ignoring the **optimization objective** itself. Consequently, a fundamental question remains unaddressed: within clipping-based RLVR objectives, which internal mechanism creates a key bottleneck for stability and convergence?

Within this scope, we perform a systematic analysis of the widely used GRPO-style clipping objective:

$$\mathcal{J}_{\text{GRPO}}(\theta) = \mathbb{E}\left[\min\left(r_t(\theta)\hat{A}_t, \underbrace{\text{clip}(r_t(\theta), 1-\epsilon, 1+\epsilon)\hat{A}_t}_{\text{Hard-Clipping Mechanism}}\right)\right],$$ (1)

[1]Shenzhen Graduate School, Peking University [2]Qwen Pilot Team [3]Dartmouth College [4]University of Toronto [5]Alibaba. Correspondence to: Guoyin Wang <guoyin.wang@alibaba-inc.com>, Li Yuan <yuanli-ece@pku.edu.cn>.

*Proceedings of the 43rd International Conference on Machine Learning*, Seoul, South Korea. PMLR 306, 2026. Copyright 2026 by the author(s).

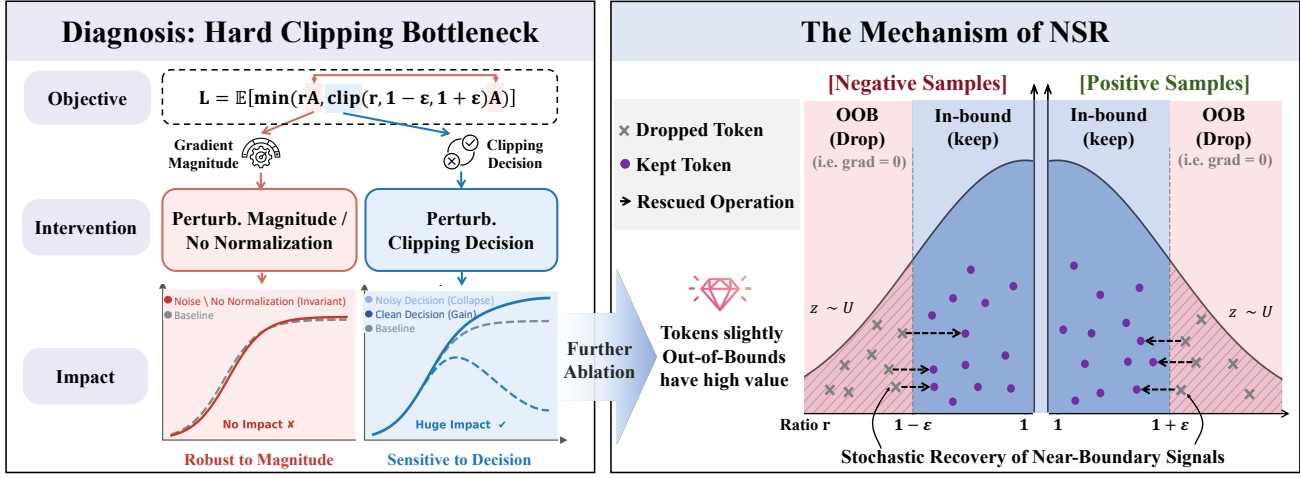

*Figure 1.* **Overview of Diagnosis and the NSR Solution. Left (Diagnosis):** Controlled interventions reveal that training is robust to gradient magnitude (red) but hypersensitive to the binary clipping decision (blue), identifying the rigid discarding of boundary signals as a key bottleneck in the studied clipping-based setup. **Right (Mechanism):** NSR mitigates this by stochastically rescuing tokens that fall slightly outside the trust region, transforming the rigid binary decision into a probabilistic admission process that retains informative near-boundary signals.

where $r_t(\theta)$ is the importance ratio and $\hat{A}_t$ is the advantage. Mechanically, the highlighted hard-clipping operation creates a rigid gradient cutoff: any token falling outside the trust region is **detached** from the computation graph, contributing zero gradient to the policy update.

Through a systematic dissection of the GRPO-style clipping objective, we identify this strict *clipping decision* as a key source of instability and signal loss in the studied setups (diagnostic process visualized in the Left panel of Figure 1). While intended to shield the model from overly large updates, this binary keep/drop mechanism is indiscriminate regarding the degree of violation—it fails to distinguish between severe divergences and marginal shifts. Consequently, it can discard informative learning signals located in the near-boundary region just beyond the clipping threshold.

Motivated by this diagnosis, we propose **Near-boundary Stochastic Rescue (NSR)**, a minimal fix focused on the clipping boundary. For tokens falling into this near-boundary region, NSR avoids indiscriminately eliminating their gradients. Instead, it employs a stochastic sampling rule to retain these tokens based on their degree of deviation from the clipping threshold (Figure 1, Right). This transforms the rigid binary decision into a probabilistic admission process, preserving potentially useful learning signals despite slight boundary violations. Crucially, NSR does not seek to indiscriminately expand the trust region; rather, it adjusts the boundary behavior, ensuring policy updates remain conservative while **retaining key gradients**.

We further provide a mechanistic interpretation of NSR. Theoretically, NSR functions in expectation as an implicit soft-clipping mechanism, imposing an approximate $1/r^2$

gradient decay on out-of-bounds tokens. However, our ablation studies reveal that the benefits of NSR extend beyond this expectation-level smoothing: the stochastic formulation consistently outperforms deterministic soft-clipping baselines, indicating that the *probabilistic admission* near the boundary is an important driver of **optimization robustness**. Validated by extensive experiments consuming **hundreds of thousands of GPU hours** across models ranging from 7B to 30B parameters and covering both dense and MoE architectures, NSR yields consistent gains as a plug-in fix on strong clipping-based baselines such as DAPO and GSPO.

In summary, we (i) diagnose hard-clipping decisions as a key source of instability and signal loss in the studied clipping-based GRPO-style RLVR setups, (ii) propose NSR to rescue near-boundary signals stochastically, and (iii) provide an expectation-level soft-clipping interpretation alongside ablations that isolate the benefit of stochastic boundary-local rescue.

**Conflict of Interest Disclosure.** Several authors are employees of Alibaba Group, which developed the Qwen series of models used in this study. This work strictly utilizes the publicly accessible open-source weights of Qwen.

## 2. Preliminaries

This section reviews the foundational policy optimization frameworks that underpin our work: PPO and its value-network-free variants, GRPO and DAPO.

**Proximal Policy Optimization (PPO)** (Schulman et al., 2017). PPO stabilizes training by constraining policy updates within a trust region around the old policy $\pi_{\theta_{old}}$. It

maximizes a clipped surrogate objective defined as:

$$\mathcal{J}_{\text{PPO}}(\theta) = \mathbb{E}_{(q,a)\sim\mathcal{D}, o\sim\pi_{\theta_{\text{old}}}(\cdot|q)}$$
$$\left[ \min\left( r_t(\theta)\hat{A}_t, \text{clip}(r_t(\theta), 1-\epsilon, 1+\epsilon)\hat{A}_t \right) \right] \quad (2)$$

where $r_t(\theta) = \frac{\pi_\theta(o_t|q,o_{<t})}{\pi_{\theta_{\text{old}}}(o_t|q,o_{<t})}$ denotes the token-level probability ratio at step $t$, $\hat{A}_t$ is the advantage estimated via a learned value network, and $\epsilon$ is the clipping coefficient.

**Group Relative Policy Optimization (GRPO) (Shao et al., 2024).** It eliminates the dependency on a value network to reduce computational overhead. Instead, it estimates relative advantages using group-based sampling. For a query $q$, a group of outputs $\{o_i\}_{i=1}^{G}$ is sampled from the old policy. The *sequence-level* advantage for the $i$-th sample is computed as:

$$\hat{A}_i = \frac{R_i - \mu_R}{\sigma_R}, \quad \text{where } R_i = \mathbb{I}(\text{Verify}(q, o_i)). \quad (3)$$

**Decoupled Clip and Dynamic sAmpling Policy Optimization (DAPO) (Yu et al., 2025).** DAPO extends GRPO by removing the explicit KL penalty and instead employs asymmetric clipping $(1 - \epsilon_{\text{low}}, 1 + \epsilon_{\text{high}})$ to amplify updates for advantageous actions, coupled with token-level advantage normalization. Furthermore, it enforces a dynamic sampling constraint that guarantees a mix of positive and negative samples within each group for stability. We adopt DAPO as the primary baseline for this work.

**Implementation Note: The Mechanics of Hard Clipping.** In practical implementations of RLVR algorithms for LLMs, the theoretical clip operator is universally instantiated via clamping functions (`torch.clamp` in PyTorch). Mechanistically, this operation creates a "hard" gradient boundary:

$$\frac{\partial \text{clip}(r_t, l, u)}{\partial r_t} = \mathbb{I}(l \le r_t \le u). \quad (4)$$

Consequently, whenever $r_t$ falls outside the trust region, its gradient is strictly zeroed out, effectively *detaching* the token and discarding its learning signal.

## 3. Empirical Diagnosis

To precisely identify the root cause of instability and suboptimal convergence in RLVR, we conduct a set of causal interventions on two core aspects of the GRPO-style objective: (i) the **magnitude** of token-level gradients and (ii) the binary **decision** induced by hard clipping.

### 3.1. Magnitude is Not the Culprit

Many existing improvements employ re-weighting or normalization to stabilize training (Wang et al., 2025c; Liu et al.,

2025a; Xu & Ding, 2025), implicitly assuming that instability and suboptimal convergence in RLVR stem from the numerical scale of advantages or importance ratios. To test this hypothesis, we design two counterfactual interventions on the standard DAPO baseline:

- **Removing Advantage Normalization (w/o Norm; A):** We remove DAPO's default advantage normalization and directly use the raw binary rewards as advantage estimates, i.e., $\hat{A}_t = R_t \in \{1, -1\}$.
- **Advantage Scalar Noise (+ Adv. Noise; B):** We inject multiplicative noise into the advantage estimate: $\hat{A}_{\text{perturb}} = \hat{A}_t \cdot z$, where $z \sim \mathcal{U}(0.8, 1.2)$. Crucially, this affects only the update magnitude, leaving the clipping decision (based on $r_t$) unchanged.

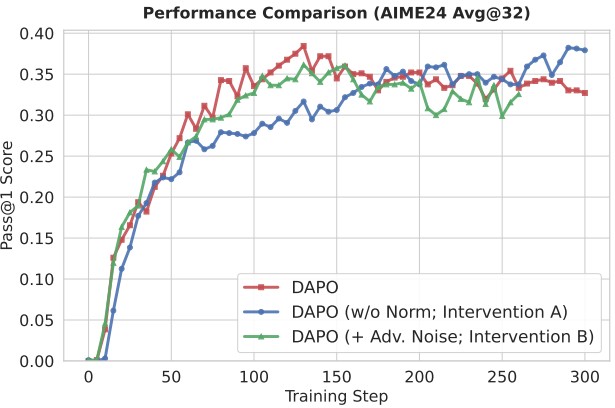

*Figure 2.* **Magnitude Insensitivity.** Comparisons between standard DAPO and magnitude-focused interventions on AIME24. Neither removing advantage normalization (*w/o Norm*) nor applying scalar multiplicative perturbation to the advantage (*+ Adv. Noise*) significantly deviates from the baseline trajectory.

To ensure the reliability of our diagnosis, all experiments are repeated across three independent runs. We compare these interventions against DAPO under identical training settings. As shown in Figure 2, while the training trajectories exhibit minor fluctuations, the **peak performance remains** robust across all settings. Standard DAPO achieves a peak Pass@1 of $\approx 37\%$ (step 140), comparable to $\approx 38\%$ for Intervention A (step 280) and $\approx 36.8\%$ for Intervention B (step 140). This counterintuitive result suggests that variance in gradient magnitude does not compromise the optimal solution, implying that numerical scale is unlikely to be the primary cause of suboptimal convergence.

### 3.2. Formalization: Decoupling Decision and Execution

Since gradient magnitude is not the bottleneck, we focus on the discrete mechanism: the **binary clipping decision**. To isolate its impact, we mathematically decouple the importance ratio's role in *adjudication* versus *optimization*.

**Trust Region Definition.** First, we define the advantage-

dependent trust region $\mathcal{I}(\hat{A}_t)$ used in standard RLVR (e.g., DAPO's asymmetric bounds):

$$\mathcal{I}(\hat{A}_t) = \begin{cases} (-\infty, 1 + \epsilon_{\text{high}}] & \text{if } \hat{A}_t > 0, \\ [1 - \epsilon_{\text{low}}, +\infty) & \text{if } \hat{A}_t < 0. \end{cases} \quad (5)$$

**Dual Roles of the Ratio.** We distinguish between two functional forms of the importance ratio derived from $r_t(\theta)$:

$$r_{\text{dec},t} \quad \text{(The Judge: determines the clipping mask)} \quad (6)$$
$$r_{\text{exec},t} \quad \text{(The Executor: drives the policy update)} \quad (7)$$

In standard algorithms, these are coupled ($r_{\text{dec}} = r_{\text{exec}} = r_t$). To analyze them separately, we introduce a perturbation $z_t$, allowing us to vary $r_{\text{exec},t} = r_{\text{dec},t} \cdot z_t$ independently.

**Decoupled Gradient Mechanics.** The hard-clipping operator in PPO / GRPO acts as a conditional gate. We can express the effective gradient contribution of a token as:

$$g_t = \underbrace{\mathbb{I}(r_{\text{dec},t} \in \mathcal{I}(\hat{A}_t))}_{\text{Binary Decision } (m_t)} \cdot \underbrace{\nabla_\theta(r_{\text{exec},t} \hat{A}_t)}_{\text{Execution Gradient}} \quad (8)$$

Here, $\mathcal{I}(\hat{A}_t)$ is the advantage-dependent trust region. Equation 8 highlights the causal chain: $r_{\text{dec}}$ exclusively controls the binary mask $m_t$ (whether the gradient flows), while $r_{\text{exec}}$ determines the magnitude of the update if admitted.

### 3.3. Isolating the Clipping Decision

We hypothesize that instability stems from the rigidity of the binary decision $m_t$ defined above. To test this, we utilize the formalism in Equation 8 to conduct a controlled intervention using noise $z \sim \mathcal{U}(0.8, 1.2)$:

- **Coupled Ratio Noise (Noisy Decision; C):** We perturb both roles simultaneously ($r_{\text{dec}} = r_{\text{exec}} = r_t \cdot z$). This conflates the clipping mask with the gradient value, allowing noise to erroneously flip the mask $m_t$.
- **Decoupled Ratio Noise (Clean Decision; D):** We perturb only the execution value ($r_{\text{exec}} = r_t \cdot z$) while keeping the decision clean ($r_{\text{dec}} = r_t$). This ensures the original signal determines the mask $m_t$, while noise modulates only the update magnitude.

We emphasize that this intervention is a diagnostic probe rather than the proposed training rule. Its purpose is to causally separate the clipping decision from the execution magnitude.

The results in Figure 3 reveal a stark divergence. **Noisy Decision (C)** leads to significant instability and entropy collapse. Mechanistically, this occurs because random perturbations systematically flip the keep/drop decision near the boundary—erroneously masking in-bound updates or admitting out-of-bound ones. Conversely, **Clean Decision**

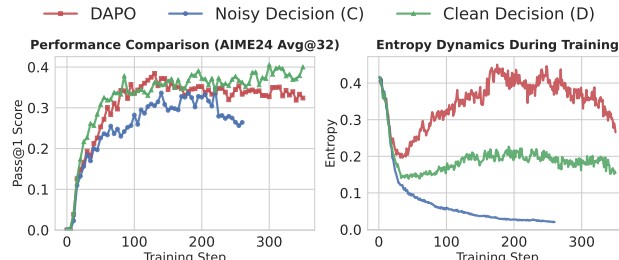

*Figure 3.* **Decision Sensitivity. Noisy Decision** (C) interferes with the clipping mask, leading to collapse. In contrast, **Clean Decision** (D) preserves the decision while perturbing gradients, which stabilizes training and outperforms the baseline.

**(D)** stabilizes training and delivers consistent performance gains (e.g. $\approx 40\%$ vs $37\%$ baseline), despite the presence of comparably scaled noise. The fact that identical noise triggers collapse in (C) but yields gains in (D) isolates the *alteration of the clipping decision* as the key causal factor in this controlled comparison.

> **TAKEAWAY**
>
> **The Decision is a Key Bottleneck.** In the studied clipping-based RLVR setup, optimization is robust to gradient magnitude but hypersensitive to the clipping decision. Instability arises when the binary verdict changes, identifying the hard boundary as a rigid bottleneck that can discard valuable signals near the trust region.

## 4. Mechanism Analysis: Where the Gains Come From

The effect of **Decoupled Ratio Noise** (D) implies that the performance gain arises from the discrepancy between the decision ratio ($r_{\text{dec}}$) and the execution ratio ($r_{\text{exec}}$). In this section, we demonstrate this mechanism through visualization and targeted ablation studies to pinpoint the exact source of improvement.

### 4.1. Visualizing Boundary Misclassification

To understand the impact of the decoupled noise intervention (Sec. 3.3), we examine the relationship between the decision ratio $r_{\text{dec}}$ and the execution ratio $r_{\text{exec}}$.

Under standard training, these ratios are strictly coupled ($r_{\text{dec}} = r_{\text{exec}}$), confining all token updates to the diagonal. However, noise $z$ breaks this constraint ($r_{\text{exec}} = r_{\text{dec}} \cdot z$), scattering the updates into a two-dimensional space as visualized in Figure 4. By mapping the trust region boundaries onto this 2D space, we categorize every token update into three distinct zones based on the coordinates ($r_{\text{dec}}, r_{\text{exec}}$):

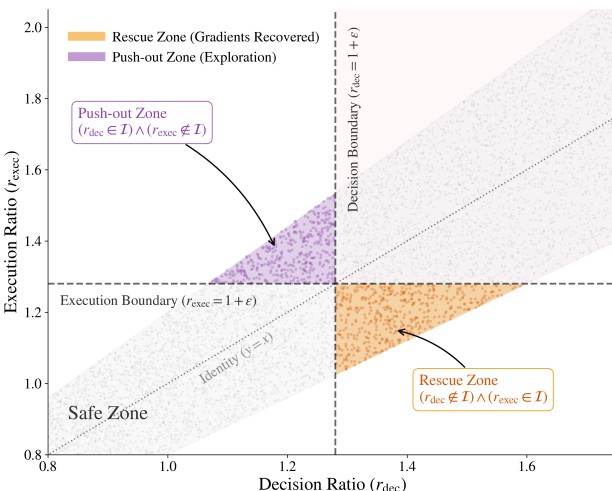

*Figure 4.* Two-dimensional landscape of Decision ($r_{\text{dec}}$) vs. Execution ($r_{\text{exec}}$). **Rescue Zone (Yellow):** Out-of-bound tokens ($r_{\text{dec}} \notin \mathcal{I}$) are recovered when noise pulls their execution value back into the trust region. **Push-out Zone (Purple):** In-bound tokens are pushed out ($r_{\text{exec}} \notin \mathcal{I}$), effectively exaggerating their update magnitude.

- **Safe Zone ($r_{\textbf{dec}}, r_{\textbf{exec}} \in \mathcal{I}$):** Both ratios remain within the trust region.
- **Rescue Zone ($r_{\textbf{dec}} \notin \mathcal{I}, r_{\textbf{exec}} \in \mathcal{I}$):** Originally clipped tokens are stochastically pulled back into the trust region, recovering gradients that were otherwise censored.
- **Push-out Zone ($r_{\textbf{dec}} \in \mathcal{I}, r_{\textbf{exec}} \notin \mathcal{I}$):** Originally in-bound tokens are pushed out by noise. While retained by the mask, their update magnitudes are effectively exaggerated beyond the trust region.

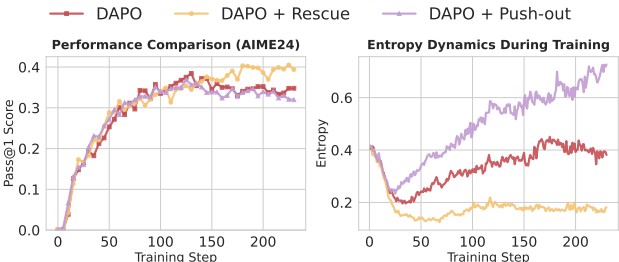

*Figure 5.* **Targeted Ablation: Rescue is the Driver.** Only-Rescue (Yellow) replicates the gains of the full decoupled method, significantly outperforming the Baseline. In contrast, **Only-Push-out** (Purple) yields no gain.

### 4.2. Targeted Ablations: Rescue vs. Push-Out

The decoupled noise intervention (Sec. 3.3) simultaneously activates both the **Rescue** and **Push-out** zones identified above. To determine which mechanism drives the performance gain, we disentangle them via conditional masking:

- **Only-Rescue (Baseline + Rescue):** We apply the de-

coupled noise mechanism *only* to tokens falling into the **Rescue Zone**. For all other tokens (including Safe and Push-out zones), we revert to the baseline behavior (standard hard clipping).
- **Only-Push-out (Baseline + Push-out):** We apply noise *only* to tokens in the **Push-out Zone**, forcing their execution values out of bounds. All other tokens follow the baseline logic.

All ablation runs are similarly repeated three times, and the results in Figure 5 show a clear pattern. **Only-Push-out** yields no significant improvement (and even slight degradation), indicating that artificially exaggerating safe updates is ineffective. Conversely, **Only-Rescue** replicates the full performance trajectory of the decoupled intervention (D). This suggests that the observed gains primarily stem from recovering informative gradients that are otherwise censored by the hard boundary.

> **TAKEAWAY**
>
> **Targeted Rescue is the Main Driver.** Our ablations suggest that performance gains primarily arise from recovering gradients in the "Rescue Zone," while the "Push-out" effect contributes little in our setup. This indicates that standard hard clipping can act as an over-aggressive filter that suppresses valuable learning signals just beyond the trust region.

## 5. A Minimal Fix: NSR

Building on the diagnosis in Section 4, we propose **Near-boundary Stochastic Rescue (NSR)**. NSR operationalizes the *Only-Rescue* principle: it strictly adheres to the baseline decision logic for in-bound tokens, while selectively attempting to recover gradients for out-of-bound tokens by stochastically mapping them back into the trust region.

### 5.1. The Algorithm

Building on the decoupled formalism in Section 3.2, NSR aims to recover gradients in the *Rescue Zone* identified in Section 4. We maintain the **Judge** as the original ratio ($r_{\text{dec},t} = r_t$) to preserve the trust-region semantics. However, we define a stochastic **Executor** that conditionally overrides the hard boundary. Specifically, we sample $z_t \sim \mathcal{U}(1-\delta, 1+\delta)$ and set $r_{\text{exec},t} = r_{\text{dec},t} \cdot z_t$.

Here, $r_t$ denotes the original importance ratio, $r_{\text{dec},t} = r_t$ is the decision ratio used for the clipping judgment, $r_{\text{exec},t} = r_{\text{dec},t} \cdot z_t$ is the stochastic candidate execution ratio, and $\tilde{r}_t$ is the final effective ratio used in the surrogate objective. The effective ratio used for optimization is:

$$\tilde{r}_t = \begin{cases} r_{\text{dec},t} & \text{if } r_{\text{dec},t} \in \mathcal{I} \\ r_{\text{exec},t} & \text{if } r_{\text{dec},t} \notin \mathcal{I} \wedge r_{\text{exec},t} \in \mathcal{I} \\ \text{clip}(r_{\text{dec},t}, \mathcal{I}) & \text{otherwise.} \end{cases} \quad (9)$$

Equation 9 highlights the **surgical nature** of NSR. Case 1 preserves the baseline behavior for in-bound tokens. Case 2 is the only rescue case: an originally out-of-bound token is admitted only when its stochastic execution ratio falls back into the trust region. Case 3 falls back to standard clipping for tokens that remain out of bounds after the rescue attempt. Thus, NSR is not a global relaxation of the trust region, but a conditional intervention that **rescues exclusively on out-of-bound tokens** near the boundary. This makes NSR a boundary-local, plug-and-play modification with negligible computational overhead.

### 5.2. Experiments

#### 5.2.1. EXPERIMENTAL SETUP.

We evaluate NSR on standard math benchmarks (AIME24/25, AMC) and general tasks (GPQA, MMLU-Pro) across three model scales: Qwen2.5-Math-7B (Dense), Qwen3-8B (Dense), and Qwen3-30B (MoE). All evaluations are zero-shot ($k = 32, T = 1.0$). We compare NSR against DAPO, keeping all hyperparameters identical except for the rescue mechanism. By default, NSR uses a rescue window of $\delta = 0.1$ (i.e., $z \sim \mathcal{U}(0.9, 1.1)$). Full hyperparameters are detailed in Appendix.

#### 5.2.2. MAIN RESULTS.

**Consistent gains across domains and architectures.** Table 1 and Table 2 summarize the performance. On math benchmarks (Table 1), NSR delivers consistent Pass@1 and pass@16 improvements across almost all models, notably boosting the 7B model by over an average of 10%. **On general reasoning** (Table 2), NSR yields significant gains on the STEM-focused GPQA dataset and maintains competitive performance on MMLU-Pro, demonstrating that the benefits of stochastic rescue generalize robustly beyond math tasks.

**Training Dynamics and Stability.** As shown in Figure 6, NSR yields significantly more stable optimization trajectories than the baseline. **(Left)** Validation accuracy converges faster and reaches a higher peak. **(Center)** Policy entropy remains stable, successfully avoiding the collapse patterns observed in DAPO. **(Right)** Response length exhibits steady growth, suggesting that rescued near-boundary gradients encourage the model to develop deeper reasoning chains.

**Mechanism and Universality.** NSR recovers near-boundary signals to effectively lower the clip fraction (see Appendix Figure 10), directly enhancing sample efficiency. As a plug-and-play solution, NSR generalizes beyond token-level clipping (DAPO) to sequence-level clipping (GSPO), where it adapts to rescue entire discarded sequences, demonstrating robustness across optimization granularities.

## 6. Theory and Validation

NSR is a minimal operator: stochastic near-boundary admission. We ask: (i) why does it stabilize RLVR, and (ii) why is it more effective than deterministic gradient decay? We answer (ii) via layered ablations (signal existence, softness, stochastic robustness), and answer (i) via an expectation-level interpretation in which NSR induces an implicit soft-clipping–like profile with $\approx O(1/r^2)$ gradient attenuation.

### 6.1. Theoretical Derivation

We omit the step index $t$ and derive the case $\hat{A} > 0$, where the active constraint is the upper bound $u$ (e.g., $u = 1 + \epsilon$). The case $\hat{A} < 0$ is symmetric with the lower bound $l$ (visualized in Figure 8 in Appendix A.4).

In NSR, the stochastic rescue process transforms the original hard-clipped ratio into a random variable $\tilde{r}(r)$:

$$\tilde{r}(r) = \begin{cases} r & \text{if } r \leq u \quad \text{(Safe)} \\ \min(r \cdot z, u) & \text{if } r > u \quad \text{(Rescue Attempt)} \end{cases} \quad (10)$$

where $z \sim U(1 - \delta, 1 + \delta)$. For $r > u$, NSR rescues the token if $rz \leq u$ (using $\tilde{r} = rz$); otherwise $\tilde{r} = u$ and the boundary contributes zero gradient.

**Proposition 6.1** (Expected Effective Ratio). *The expected objective of NSR, $f(r) \triangleq \mathbb{E}_z[\tilde{r}(r)]$, admits the following closed-form solution. For the rescue zone ($u < r < \frac{u}{1-\delta}$):*

$$f(r) = \frac{1}{2\delta}\left(u(1 + \delta) - \frac{u^2}{2r} - \frac{(1-\delta)^2 r}{2}\right). \quad (11)$$

*For deep violations ($r \geq \frac{u}{1-\delta}$), the expectation saturates to the constant $u$.*

Differentiation of $f(r)$ yields the effective gradient scaling:

**Corollary 6.2** (Gradient Decay Profile). *Within the rescue zone ($u < r < \frac{u}{1-\delta}$), the expected gradient of NSR, $g(r) \triangleq \frac{\mathrm{d}}{\mathrm{d}r} f(r)$, follows:*

$$g(r) = \frac{1}{4\delta}\left(\frac{u^2}{r^2} - (1-\delta)^2\right). \quad (12)$$

**Mechanism Interpretation.** Corollary 6.2 reveals that NSR replaces the binary censorship of hard clipping with a continuous weighting scheme. As visualized in Figure 7, the gradient decays following an inverse-square law ($\approx O(1/r^2)$).

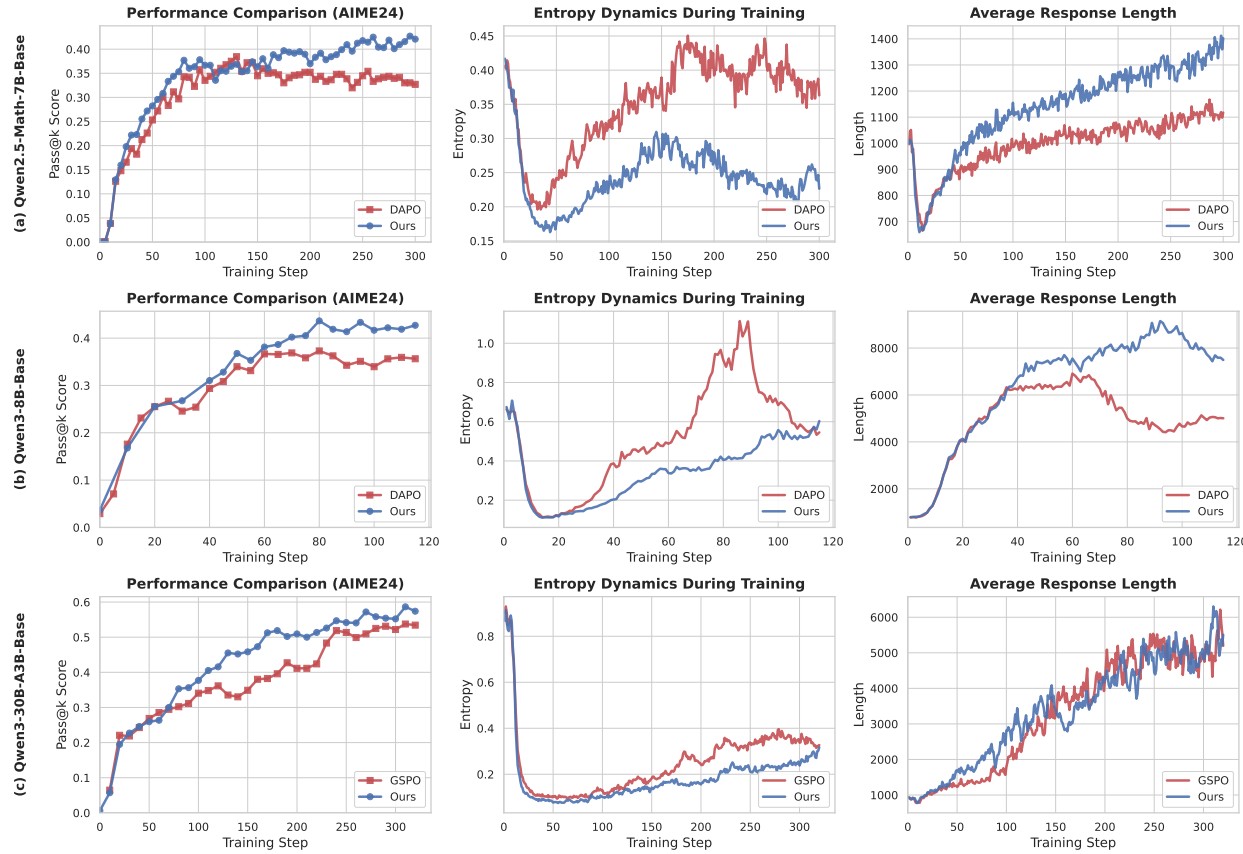

*Figure 6.* **Training dynamics.** NSR exhibits more stable training than DAPO. **Left:** validation performance converges faster and to a higher value. **Center:** policy entropy remains stable, avoiding the collapse commonly seen in the baseline. **Right:** response length increases steadily, consistent with rescued near-boundary gradients promoting longer, more complete reasoning chains.

*Table 1.* **Main results on math reasoning benchmarks.** Comparison of NSR with strong baselines (DAPO and GSPO) across model scales and architectures. Red denotes an improvement and blue a decline.

| Model | Setting | Method | AIME24 | | AIME25 | | AMC | |
|---|---|---|---|---|---|---|---|---|
| | | | Pass@1 | Pass@16 | Pass@1 | Pass@16 | Pass@1 | Pass@16 |
| Qwen2.5-Math-7B | Dense + 8k | DAPO | 35.83 | 55.05 | 15.90 | 30.70 | 69.28 | 88.24 |
| | | + NSR | $40.76^{+4.93}$ | $57.70^{+2.65}$ | $17.57^{+1.67}$ | $32.78^{+2.08}$ | $76.74^{+7.46}$ | $89.36^{+1.12}$ |
| Qwen3-8B-Base | Dense + 30k | DAPO | 37.29 | 75.18 | 27.39 | 48.98 | 71.42 | 88.52 |
| | | + NSR | $43.65^{+6.36}$ | $75.39^{+0.21}$ | $31.67^{+4.28}$ | $55.87^{+6.89}$ | $74.92^{+3.50}$ | $91.28^{+2.76}$ |
| Qwen3-30B-A3B-Base | MoE + 20k | GSPO | 54.17 | 79.74 | 38.44 | 56.67 | 81.66 | 92.01 |
| | | + NSR | $58.65^{+4.48}$ | $83.26^{+3.52}$ | $40.53^{+2.09}$ | $64.83^{+8.16}$ | $85.09^{+3.43}$ | $94.37^{+2.36}$ |

This decay arises from two coupled effects: (1) **Decreasing Admission Probability** ($p_{\text{rescue}} \propto 1/r$), and (2) **Magnitude Modulation** (larger $r$ requires smaller $z$ to be admitted). This "soft constraint" applies symmetrically to both upper and lower boundaries (see Appendix Figure 9 for the lower-bound profile), recovering valuable near-boundary signals while conservatively suppressing deep deviations.

### 6.2. Empirical Validation: A Layered Ablation

Our theoretical analysis establishes that NSR induces an implicit gradient decay (approx. $\propto 1/r^2$) in expectation ( Corollary 6.2). This raises a fundamental question: **does the performance gain stem merely from this attenuation profile, or is the stochastic nature of the rescue essential?**

To answer this and address why one cannot simply apply a deterministic decay, we conduct a layered ablation that

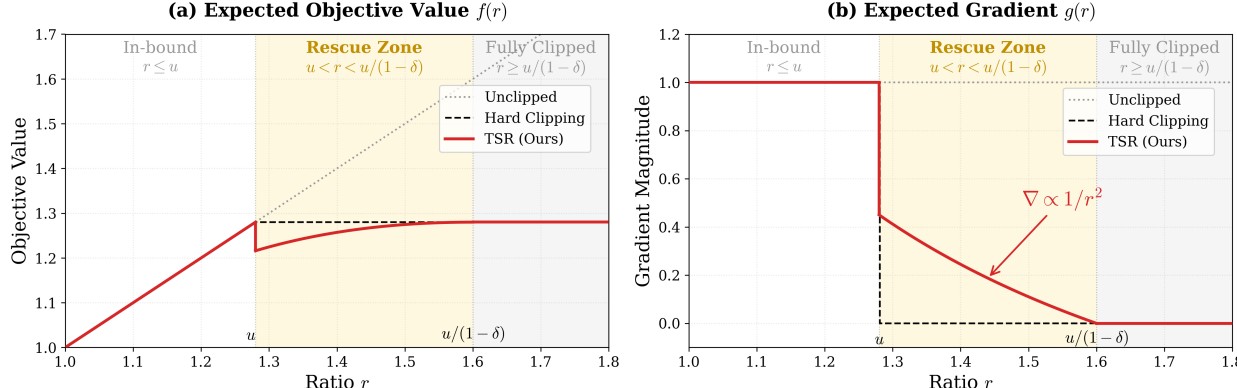

*Figure 7.* **NSR induces implicit soft clipping in expectation. Left:** The expected effective ratio $f(r)$ transitions from the linear identity to a smooth saturation curve within the rescue zone ($u < r < \frac{u}{1-\delta}$). **Right:** The corresponding expected gradient $g(r)$ replaces the binary gate of hard clipping with a smooth decay dominated by the $O(1/r^2)$ term in Equation 12.

*Table 2.* Results on general science benchmarks (GPQA and MMLU-Pro (Wang et al., 2024)).

| Model | Method | GPQA | MMLU-Pro |
|---|---|---|---|
| Qwen2.5-Math-7B | DAPO
+ NSR | 39.27
$42.99^{+3.72}$ | 44.09
$46.45^{+2.36}$ |
| Qwen3-8B-Base | DAPO
+ NSR | 52.02
$53.16^{+1.14}$ | 65.67
$66.41^{+0.74}$ |
| Qwen3-30B-A3B-Base | GSPO
+ NSR | 58.96
$59.22^{+0.26}$ | 72.08
$74.27^{+2.19}$ |

disentangles **signal admission** (existence), **magnitude modulation** (shape), and **stochasticity** (robustness). In this analysis, let $r$ denote the clean decision ratio $r_{dec}$ and $u$ the active trust-region bound (e.g., $1 + \epsilon_{high}$). We compare:

- **Baseline (Hard Clipping):** DAPO, which strictly zeroes out gradients when $r \notin \mathcal{I}$.

- **Binary Admission (No Modulation):** Uses perturbation $z$ solely to determine *whether* to admit an out-of-bound token (i.e., if $r \cdot z$ falls back into $\mathcal{I}$). Upon admission, it uses the boundary value (effectively $\tilde{r} = u$).

- **Explicit Soft Decay (Deterministic):** Removes stochasticity and applies a deterministic weight $w(r)$ to all out-of-bound tokens: $g_{soft}(r) = w(r)\, g_{unclipped}(r)$ with $w(r) \propto (u/r)^k$ for $k \in \{2, 3, 4\}$ (continuous at $r = u$).

**1. Admission suggests signal existence.** Compared to the Baseline, **Binary Admission** yields consistent performance gains (Table 3, Row 2) even without magnitude modulation. This provides direct evidence that useful learning signals can exist outside the clipping boundary in our studied setup; standard hard clipping may therefore introduce false negatives by censoring informative near-boundary updates.

**2. Soft Decay is beneficial but insufficient.** Explicit **Soft Decay** (deterministic, $w(r) = (u/r)^k$) consistently improves over the Baseline across $k \in \{2, 3, 4\}$ (Table 3, Rows 3–5), indicating that smooth, conservative attenuation is an important ingredient for stability. However, the performance–stability trade-off depends on the decay shape (choice of $k$), and even the best deterministic decay still falls short of NSR in aggregate stability (especially in Pass@16).

**3. A plausible mechanism: stochastic filtering vs. deterministic attenuation.** While **Explicit Soft Decay** improves over the Baseline, Table 3 shows larger run-to-run variance than NSR, suggesting that deterministic shrinking may remain sensitive to noisy out-of-bound directions. In contrast, **NSR (Ours)** delivers better aggregate consistency and performance. We view this as evidence for a plausible mechanism rather than a formal proof of superiority: deterministic decay keeps every slightly out-of-bound direction persistently available, albeit with a smaller weight, whereas NSR admits such directions only probabilistically and distance-dependently. In this sense, NSR behaves like a boundary-local stochastic filter. Useful near-boundary signals can be recovered in expectation, while transient or unreliable out-of-bound directions are sometimes rejected entirely instead of being repeatedly propagated with small deterministic weights. This filtering interpretation is consistent with the empirical stability of NSR, but a formal bias–variance or convergence analysis is left for future work.

## 7. Related Work

**RLVR for Reasoning Scaling.** (Guo et al., 2025; Jaech et al., 2024; Ma et al., 2026; Yang et al., 2026) Following the success of OpenAI's o1 (Jaech et al., 2024) and DeepSeek-R1 (Guo et al., 2025), RLVR has become the standard paradigm for scaling reasoning. While Group Relative Policy Optimization (GRPO) (Shao et al., 2024) serves

*Table 3.* **Stability analysis across independent runs.** We report the performance of each individual run and the aggregate Mean ± Standard Deviation (SD). While individual runs show stochastic fluctuations (e.g., explicit decay occasionally peaks), **NSR** demonstrates the best stability and highest aggregate performance.

| Method | Run 1 | | Run 2 | | Run 3 | | Mean ± SD | |
|---|---|---|---|---|---|---|---|---|
| | Pass@1 | Pass@16 | Pass@1 | Pass@16 | Pass@1 | Pass@16 | Pass@1 | Pass@16 |
| DAPO (Baseline) | 37.19 | 58.00 | 35.10 | 51.48 | 35.21 | 55.68 | $35.83_{\pm1.18}$ | $55.05_{\pm3.30}$ |
| Binary Thresholding | 37.70 | 57.72 | 39.58 | 56.64 | 36.25 | **58.53** | $37.84_{\pm1.67}$ | $57.63_{\pm0.95}$ |
| Explicit Decay ($k=2$) | 40.52 | 57.20 | 36.35 | **57.27** | 39.17 | 54.23 | $38.68_{\pm2.13}$ | $56.23_{\pm1.73}$ |
| Explicit Decay ($k=3$) | 39.90 | 56.04 | **39.89** | 53.95 | 37.08 | 57.24 | $38.96_{\pm1.62}$ | $55.74_{\pm1.66}$ |
| Explicit Decay ($k=4$) | 38.85 | 50.16 | 39.06 | 55.15 | 35.31 | 56.92 | $37.74_{\pm2.11}$ | $54.08_{\pm3.50}$ |
| **NSR (Ours)** | **42.50** | **58.16** | 39.48 | 56.75 | **40.31** | 58.19 | $\mathbf{40.76_{\pm1.56}}$ | $\mathbf{57.70_{\pm0.82}}$ |

as a foundation, recent variants aim to improve its stability and efficiency. Methods like DAPO (Yu et al., 2025), GSPO (Zheng et al., 2025), and CISPO (Chen et al., 2025) refine sampling strategies and gradient estimation to mitigate training collapse (Hu et al., 2025; Zeng et al., 2025; Yue et al., 2025). In this work, we adopt DAPO—a strong open-source algorithm—as our primary baseline.

**Token-level Importance and Clipping.** Training stability in clipping-based RLVR can be strongly affected by the "hard boundary" effects inherent in policy constraint mechanisms (Schulman et al., 2015). The standard hard clipping in PPO (Schulman et al., 2017) and GRPO imposes a rigid binary verdict that can suppress informative learning signals when they fall slightly outside the trust region. To address this discontinuity, recent works have proposed explicit boundary softening techniques, such as adaptive gating (SAPO (Gao et al., 2025)), asymmetric sampling (ASPO (Wang et al., 2025a), BAPO (Xi et al., 2025)), or probability smoothing (PSPO (Dwyer et al., 2025), FSPO (Mao et al., 2025)). Unlike explicit soft clipping, NSR employs a stochastic rescue mechanism to retain informative near-boundary signals.

## 8. Conclusion

We identify the rigid binary decision of hard clipping as a key bottleneck in the clipping-based RLVR settings studied in this work, where informative near-boundary signals can be discarded by the standard hard-clipping rule. To address this, we propose Near-boundary Stochastic Rescue (NSR), a minimal boundary-local modification that stochastically rescues slightly out-of-bound updates while preserving conservative policy constraints. While NSR is theoretically interpretable as implicit soft clipping with $O(1/r^2)$ decay in expectation, our ablations suggest that its effectiveness comes from combining signal recovery with boundary-local stochastic filtering. Overall, NSR provides a simple plug-and-play mechanism for improving the stability of clipping-based RLVR training.

## Limitations and Future Work

This work is scoped to clipping-based RLVR objectives with verifiable rewards and group-relative advantage estimation. We do not claim direct transfer to standard RL with learned value functions, dense rewards, or unclipped/KL-only objectives. NSR introduces one additional hyperparameter, the rescue window $\delta$, and broader tuning guidance across clip widths and algorithms remains future work. Our experiments are also limited to Qwen-family models. Finally, the interaction between NSR, MoE routing, and token-level versus sequence-level clipping remains underexplored and deserves further study.

## Acknowledgment

This work was supported in part by the Guangdong Grants (Grant No. 2023ZT10X075), the Natural Science Foundation of China (No. 62332002, 62425101), and the Shenzhen Science and Technology Program (KQTD20240729102051063). It was also supported by the China Postdoctoral Science Foundation under Grant Numbers BX20240013 and 2024M760113.

## Impact Statement

This paper presents work whose goal is to advance the field of Machine Learning, specifically by improving the stability and efficiency of Reinforcement Learning with Verifiable Rewards (RLVR). While advancing LLM reasoning capabilities carries general societal implications regarding automation and dual-use risks, our method is a generic optimization technique designed to make existing training pipelines more robust. There are no specific ethical consequences of our work that we feel must be highlighted here beyond standard considerations for Large Language Model development.

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

# A. Proofs

This appendix provides proofs for Proposition 6.1 and Corollary 6.2 in Section 6. We follow the notation in the main text and consider the case of positive advantage ($\hat{A} > 0$), where the active trust-region constraint is the upper bound $u$. Recall that NSR defines the effective ratio random variable

$$\tilde{r}(r) = \begin{cases} r, & r \leq u \quad \text{(Safe)}, \\ \min(rz, u), & r > u \quad \text{(Rescue Attempt)}, \end{cases} \qquad z \sim \mathcal{U}(1 - \delta, 1 + \delta), \tag{13}$$

where $\delta \in (0, 1)$ and the density of $z$ is $p(z) = \frac{1}{2\delta}$ on $[1 - \delta, 1 + \delta]$.

**A note on symmetry ($\hat{A} < 0$).** For negative advantage, the same derivation applies with the lower bound $l$ as the active constraint (i.e., rescue applies when $r < l$), by replacing the upper-threshold event $\{rz \leq u\}$ with the lower-threshold event $\{rz \geq l\}$ and reversing the relevant inequalities. We omit the symmetric steps for brevity.

## A.1. Proof of Proposition 6.1

*Proof.* Define $a \triangleq 1 - \delta$ and $b \triangleq 1 + \delta$. For $r > u$, introduce the critical value

$$z_c(r) \triangleq \frac{u}{r}. \tag{14}$$

Then for any fixed $r > u$ and any $z \in [a, b]$,

$$\tilde{r}(r) = \min(rz, u) = \begin{cases} rz, & z \leq z_c(r), \\ u, & z > z_c(r). \end{cases}$$

**Step 1: piecewise regimes.** The location of $z_c(r)$ relative to $[a, b]$ yields three regimes:

- **Safe region** ($r \leq u$): $\tilde{r}(r) = r$ deterministically, hence $f(r) \triangleq \mathbb{E}_z[\tilde{r}(r)] = r$.

- **Rescue zone** ($u < r < \frac{u}{1-\delta}$): equivalently $a < z_c(r) < 1$ and in particular $a < z_c(r) < b$.

- **Deep violations** ($r \geq \frac{u}{1-\delta}$): equivalently $z_c(r) \leq a$, so $rz \geq ra \geq u$ for all $z \in [a, b]$ and thus $\tilde{r}(r) = u$ deterministically, giving $f(r) = u$.

**Step 2: compute $f(r)$ in the rescue zone.** Assume $u < r < \frac{u}{1-\delta}$ so that $z_c(r) \in (a, b)$. Using the density $p(z) = \frac{1}{2\delta}$ and the above partition,

$$f(r) = \mathbb{E}_z[\tilde{r}(r)] = \frac{1}{2\delta} \left( \int_a^{z_c(r)} rz \, \mathrm{d}z + \int_{z_c(r)}^b u \, \mathrm{d}z \right)$$
$$= \frac{1}{2\delta} \left( r \cdot \frac{z_c(r)^2 - a^2}{2} + u \cdot (b - z_c(r)) \right). \tag{15}$$

Substituting $z_c(r) = u/r$ and $(a, b) = (1 - \delta, 1 + \delta)$ into Equation 15 yields

$$f(r) = \frac{1}{2\delta} \left( \frac{r}{2} \left( \frac{u^2}{r^2} - a^2 \right) + u \left( b - \frac{u}{r} \right) \right)$$
$$= \frac{1}{2\delta} \left( ub - \frac{u^2}{2r} - \frac{a^2 r}{2} \right)$$
$$= \frac{1}{2\delta} \left( u(1 + \delta) - \frac{u^2}{2r} - \frac{(1 - \delta)^2 r}{2} \right),$$

which matches Proposition 6.1. The saturation statement for $r \geq \frac{u}{1-\delta}$ follows from the "Deep violations" case above. $\square$

## A.2. Proof of Corollary 6.2

*Proof.* We prove the stated formula for the rescue zone $u < r < \frac{u}{1-\delta}$. From Proposition 6.1, in this interval

$$f(r) = \frac{1}{2\delta}\left(u(1+\delta) - \frac{u^2}{2r} - \frac{(1-\delta)^2 r}{2}\right).$$

Differentiating w.r.t. $r$ gives

$$\begin{aligned} g(r) \triangleq \frac{\mathrm{d}}{\mathrm{d}r}f(r) &= \frac{1}{2\delta}\left(0 - \frac{u^2}{2}\cdot(-r^{-2}) - \frac{(1-\delta)^2}{2}\right) \\ &= \frac{1}{4\delta}\left(\frac{u^2}{r^2} - (1-\delta)^2\right), \end{aligned}$$

which is exactly Equation 12.

**Equivalent "differentiate-under-expectation" view (optional).** Define $\phi(r,z) \triangleq \min(rz, u)$ for $r > u$. For each fixed $z \in [a,b]$, $\phi(r,z)$ is differentiable in $r$ for all $r \neq u/z$, with

$$\frac{\partial}{\partial r}\phi(r,z) = z \cdot \mathbf{1}\{rz < u\},$$

and the non-differentiable points $r = u/z$ form a measure-zero set under $z$. Moreover, $\left|\frac{\partial}{\partial r}\phi(r,z)\right| \leq b$ is integrably bounded. Thus, by a standard dominated-convergence / Leibniz argument, we may interchange derivative and expectation almost everywhere:

$$\frac{\mathrm{d}}{\mathrm{d}r}\mathbb{E}_z[\phi(r,z)] = \mathbb{E}_z\left[\frac{\partial}{\partial r}\phi(r,z)\right] = \frac{1}{2\delta}\int_a^{z_c(r)} z\,\mathrm{d}z = \frac{1}{4\delta}\left(z_c(r)^2 - a^2\right) = \frac{1}{4\delta}\left(\frac{u^2}{r^2} - (1-\delta)^2\right).$$

This recovers the same gradient profile and makes explicit the inverse-square decay driven by the shrinking admissible interval $[a, z_c(r)]$. $\qquad\square$

## A.3. Additional piecewise forms (for completeness)

For reference, combining the regimes in Appendix A.1 yields the full piecewise expressions:

$$f(r) = \mathbb{E}_z[\tilde{r}(r)] = \begin{cases} r, & r \leq u, \\ \frac{1}{2\delta}\left(u(1+\delta) - \frac{u^2}{2r} - \frac{(1-\delta)^2 r}{2}\right), & u < r < \frac{u}{1-\delta}, \\ u, & r \geq \frac{u}{1-\delta}, \end{cases} \tag{16}$$

and

$$g(r) = \frac{\mathrm{d}}{\mathrm{d}r}f(r) = \begin{cases} 1, & r < u, \\ \frac{1}{4\delta}\left(\frac{u^2}{r^2} - (1-\delta)^2\right), & u < r < \frac{u}{1-\delta}, \\ 0, & r > \frac{u}{1-\delta}, \end{cases} \tag{17}$$

where boundary points can be defined consistently by continuity (e.g., $f(u) = u$ and $g(u^-) = 1$, $g(u^+) = \frac{1}{4\delta}(1-(1-\delta)^2)$).

## A.4. Symmetric Results for Lower Bound ($\hat{A} < 0$)

For completeness, we provide the explicit forms for the negative advantage case ($\hat{A} < 0$), where the active constraint is the lower bound $l = 1 - \epsilon$. Due to symmetry, the derivation follows Appendix A.1 and A.2 with inverted inequalities.

**Rescue Zone.** The rescue mechanism activates when the ratio drops below the lower bound ($r < l$) but remains recoverable. The symmetric rescue zone is defined as:

$$\frac{l}{1+\delta} < r < l. \tag{18}$$

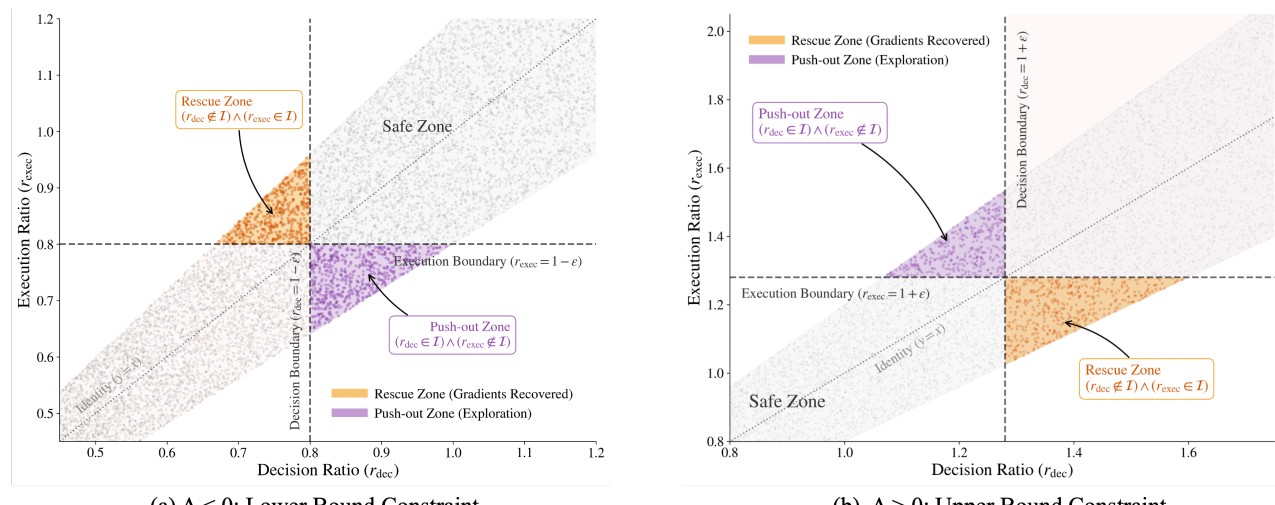

(a) A < 0: Lower Bound Constraint  (b) A > 0: Upper Bound Constraint

*Figure 8.* **Symmetric decision–execution geometry for upper and lower trust-region constraints.** We visualize token updates in the 2D space of decision ratio $r_{\text{dec}}$ (judge) versus execution ratio $r_{\text{exec}}$ (executor) under a decoupled perturbation. **Left ($\hat{A} < 0$):** the active constraint is the lower bound $l = 1 - \epsilon$; the *Rescue Zone* corresponds to $r_{\text{dec}} < l$ but $r_{\text{exec}} \geq l$, while the *Push-out Zone* corresponds to $r_{\text{dec}} \geq l$ but $r_{\text{exec}} < l$. **Right ($\hat{A} > 0$):** the active constraint is the upper bound $u = 1 + \epsilon$; the *Rescue Zone* corresponds to $r_{\text{dec}} > u$ but $r_{\text{exec}} \leq u$, and the *Push-out Zone* corresponds to $r_{\text{dec}} \leq u$ but $r_{\text{exec}} > u$. Across both cases, NSR targets the rescue region to probabilistically admit near-boundary updates while keeping deep violations clipped.

Tokens with $r \leq \frac{l}{1+\delta}$ are considered deep violations and remain fully clipped (contributing zero gradient).

**Gradient Profile.** Within the rescue zone in Equation 18, the expected gradient $g(r)$ follows the symmetric inverse-square decay:

$$g(r) = \frac{1}{4\delta}\left((1+\delta)^2 - \frac{l^2}{r^2}\right). \tag{19}$$

This confirms that NSR provides a consistent, smooth gradient attenuation mechanism for both upward and downward deviations from the trust region.

# B. Experimental Setup.

In this section, we provide comprehensive details regarding our experimental infrastructure, training configurations, and evaluation protocols to ensure reproducibility.

## B.1. Framework and Dataset

We implement all algorithms using the **VERL** framework. For the training dataset, we utilize **dapo-math-17k** across all main experiments. This choice ensures a fair and controlled comparison with the baseline methods, as it aligns with the data setting used in the original DAPO and GSPO studies.

## B.2. Model Architectures and Constraints

We evaluate our method across three representative model scales and architectures: **Qwen2.5-Math-7B** (Dense), **Qwen3-8B-Base** (Dense), and **Qwen3-30B-A3B-Base** (MoE). To accommodate the varying reasoning capabilities of these models, we enforce different maximum response length constraints during training:

- **Qwen2.5-Math-7B:** Maximum response length of **8k** tokens.

- **Qwen3-8B-Base:** Maximum response length of **30k** tokens.

- **Qwen3-30B-A3B-Base:** Maximum response length of **20k** tokens.

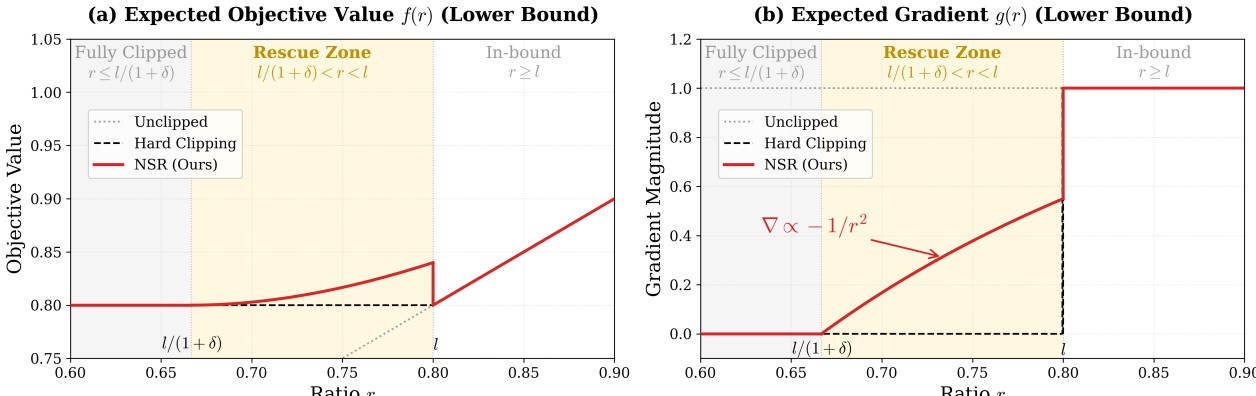

*Figure 9.* **Expectation-level soft clipping induced by NSR at the lower bound ($\hat{A} < 0$). Left:** the expected effective ratio $f(r) = \mathbb{E}[\tilde{r}(r)]$ transitions from the unclipped identity to a smooth curve inside the lower-bound rescue zone $l/(1+\delta) < r < l$, and saturates at $l$ for deep violations ($r \leq l/(1+\delta)$). **Right:** the corresponding expected gradient $g(r) = \frac{\mathrm{d}}{\mathrm{d}r} f(r)$ replaces the hard-clipping binary gate with a smooth attenuation following Equation 19.

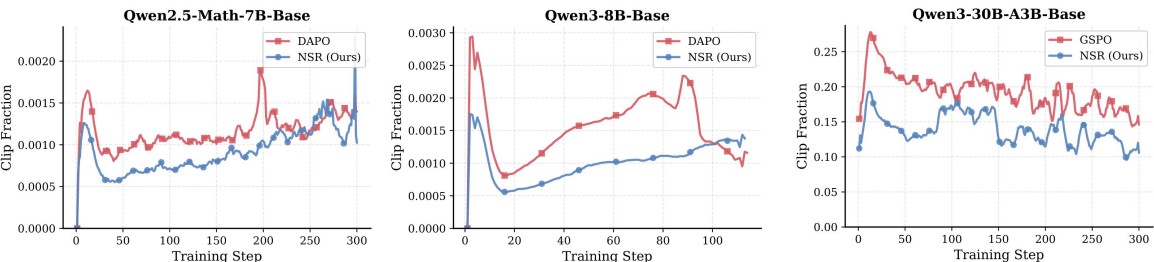

*Figure 10.* **Clip Fraction Analysis.** We monitor the fraction of clipped gradients throughout training across different model scales. **NSR (blue) consistently maintains a significantly lower clip fraction** compared to baselines (red). This reduction confirms that NSR effectively rescues valid near-boundary signals that would otherwise be discarded, thereby enhancing sample utilization. Crucially, this benefit validates the method's plug-and-play versatility: it reduces clipping not only in token-level objectives (DAPO) but also adapts to rescue entire sequences in sequence-level objectives (GSPO, bottom-right).

### B.3. Training Implementation Details

Our training configuration strictly follows the settings of the baselines (DAPO and GSPO) to isolate the contribution of our proposed method.

**General Optimization Settings.** Unless otherwise specified, we train with a global batch size of **512**. We employ gradient accumulation steps to manage memory, typically setting the mini-batch size to **32** with **16** accumulation steps. The learning rate is fixed at $10^{-6}$. Following the DAPO configuration, we exclude both KL divergence and entropy regularization losses from the objective.

**Baseline-Specific Configurations.**

- **DAPO Configuration:** We adopt the recommended hyperparameters from the DAPO literature, including clip-higher, dynamic sampling, token-level policy gradient loss, and overlong reward shaping. The clipping thresholds are set to $\epsilon_{high} = 0.28$ and $\epsilon_{low} = 0.2$.

- **GSPO Configuration (Cold-Start):** For experiments involving the GSPO algorithm, we conduct a cold-start training initialized from the *Qwen3-30B-A3B-Base* model. Aligning with the official VERL implementation scripts for GSPO, the clipping ranges are set to stringent values of $3 \times 10^{-4}$ for the lower bound and $4 \times 10^{-4}$ for the upper bound.

### B.4. NSR Hyperparameters

Our proposed Near-boundary Stochastic Rescue (NSR) introduces a perturbation range $z \sim \mathcal{U}(1 - \delta, 1 + \delta)$. The choice of $\delta$ depends on the underlying clipping mechanism of the base algorithm:

- **NSR on DAPO (Token-level Clip):** We use a default rescue window of $\delta = 0.1$, corresponding to a noise range of $z \sim \mathcal{U}(0.9, 1.1)$.

- **NSR on GSPO (Sequence-level Clip):** Given the stricter sequence-level constraints in GSPO, we apply a much narrower rescue window. We set the noise range to $z \sim \mathcal{U}(0.999, 1.001)$ (i.e., $\delta = 0.001$) to maintain stability while recovering gradients.

### B.5. Evaluation Protocol

We conduct a rigorous zero-shot evaluation across multiple benchmarks using a consistent sampling strategy with temperature $T = 1.0$ and top-$p = 0.7$.

- **Mathematical Reasoning Benchmarks (AIME 24, AIME 25, AMC):** For these datasets, we sample $k = 32$ completions for each problem and report **Pass@1** and **Pass@16**.

- **General Reasoning Benchmarks (GPQA, MMLU-Pro):** To assess generalizability, we evaluate on GPQA (STEM-focused) and MMLU-Pro. For these tasks, we sample $k = 8$ completions per problem and report the average accuracy.

## C. Computing Resources.

The training budget for each setting is detailed as follows: (1) The 7B dense model required approximately 768 GPU hours per run, covering $300 \times 16$ gradient steps. (2) The 8B dense model required approximately 6,144 GPU hours per run, covering $150 \times 16$ gradient steps. (3) The 30B MoE model required approximately 9,216 GPU hours per run, covering $450 \times 16$ gradient steps. All experiments were repeated at least three times to ensure statistical reliability.

