# OpenReview forum: "Clipping Bottleneck: Stabilizing RLVR via Stochastic Recovery of Near-Boundary Signals"
_ICML.cc/2026/Conference — ICML 2026 regular_

### Official Review · Reviewer_uHFn · 2026-03-10

**Soundness:** 3
**Presentation:** 4
**Significance:** 2
**Originality:** 3
**Overall Recommendation:** 4
**Confidence:** 4

**Summary:**

To investigate the bottlenecks that limit the stability and convergence of the standard RLVR objective, this paper conducted systematic ablation studies and identified the rigid binary decision of hard clipping as the primary bottleneck that discards high value signals. To address this, this paper proposed Near-boundary Stochastic Rescue (NSR), a minimal, plug-and-play modification that stochastically retains these slightly out-of-bound tokens to recover lost signals. Experiments across model sizes from 7B to 30B and both dense and MoE architectures validated the effectiveness of NSR.

**Compliance With Llm Reviewing Policy:**

Affirmed.

**Final Justification:**

Overall, most of my previous concerns have been addressed, with the remaining concern about the lack of comparison with methods that remove the clipping operator, to demonstrate the superiority and necessity of the proposed clipping method.

That said, I appreciate the additional experiments and empirical results presented. Accordingly, I am increasing my rating to 4.

**Key Questions For Authors:**

1.	In figure 2, 3, 4, what is the $\epsilon_{high}$ for DAPO related method?
2.	Is the rescue parameter $\delta$ a sensitive parameter?

**Limitations:**

Not discussed.

**Strengths And Weaknesses:**

Strength

1.	The paper is well organized and clearly presented, and the ablation study is systematic.

2.	Extensive experiments across model sizes from 7B to 30B are conducted to validate the proposed method.

Weakness

1.	The motivation of the method design is not clear. While in section 3.3, it’s argued that perturbation on $r_{dec}$ leads to performance degradation, the paper still proposes to perturb $r_{dec}$ with $\tilde r_{t}$ in equation (9). Besides, what is the relationship between $r'$, $r_{exec}$ and $\tilde r_{t}$  in section 5.1?

2. The advantage of the proposed method is not well validated. This paper aims to stabilize training and achieve better performance, but only the base methods DAPO and GSPO are included for comparison. There are also other works focusing on improving base methods like GRPO and DAPO. For example, [1] proposed Clip-Cov and KL-Cov to address entropy collapse and achieve
better downstream performance, [2] shows that directly removing the clipping operator(+ KL constraints) avoids entropy collapse and achieves stable entropy control. How will the proposed method perform, compared with these baselines working on the same goal?



[1] https://arxiv.org/abs/2505.22617

[2] https://arxiv.org/abs/2505.12366

---

> ### Author Rebuttal · Authors · 2026-03-30
>
> We sincerely appreciate your constructive feedback. Below we address each concern in detail.
>
> ---
>
> > W1: Clarifying the motivation and notation
>
> The motivation of Sec. 3.3 is simple: **through a perturbation-based decomposition, it shows that what should remain clean is the clipping decision, while what can be perturbed is the execution value.** This directly motivates NSR: preserve the original clipping judgment, and introduce stochasticity only on the execution side to rescue useful near-boundary signals. We will revise the text to make this design logic explicit.
>
> To make the notation in Sec. 5.1 easier to follow, we will state all variable roles explicitly and then summarize the operational cases in a compact table:
> | Quantity | Meaning |
> | --- | --- |
> | $r_t$ | original raw ratio |
> | $r_{dec,t}=r_t$ | ratio used for clipping judgment |
> | $r_{\text{exec},t}=r'_t=r_t\cdot z$ | candidate perturbed execution ratio |
> | $\tilde r_t$ | final effective ratio used in optimization |
>
> Concretely, the final effective ratio is determined as follows:
> | Condition | $\tilde r_t$ |
> | --- | --- |
> | in-bound | $r_t$ |
> | OOB but rescued | $r_{\text{exec},t}=r'_t=r_t\cdot z$ |
> | OOB and not rescued | clipped boundary value |
>
> ---
>
> > W2: Comparisons with concurrent baselines
>
> In the submission, we chose DAPO / GSPO as primary baselines because they are currently among the most commonly used and **representative open-source clipping-based RLVR baselines**, covering both **token-level and sequence-level** settings. We believe validating it first on these strong standard baselines is the most direct and informative evaluation.
>
> | Method | AIME24 | AIME25 |
> | :--- | :---: | :---: |
> | Clip-Cov | 35.83 | 16.15 |
> | + NSR | **36.67** | **18.02** |
> | KL-Cov | 34.90 | 15.52 |
> | + NSR | **36.89** | **18.44** |
>
> We also appreciate the reviewer for pointing us to concurrent work such as Clip-Cov / KL-Cov. Due to the limited rebuttal time, we additionally implemented this baseline and evaluated NSR on top of it. In our matched setup, adding NSR on Clip-Cov / KL-Cov still brings further gains. This suggests that stochastic boundary rescue is complementary and can also benefit other clipping-based stabilization methods.
>
> ---
>
> > Q1: $\epsilon_{high}$ for DAPO
>
> We already introduced DAPO’s **asymmetric clipping rule**[1] in the Preliminaries, where $\epsilon_{high}$ is intentionally set larger than $\epsilon_{low}$. For the DAPO-related results in Figs. 2, 3, and 4, we use:
> $$\epsilon_{high}=0.28,\qquad \epsilon_{low}=0.2$$
> These settings are also provided in **Appendix B.3**. We will add them directly to the relevant figure captions and/or main text in the revision for clarity.
>
> ---
>
> > Q2:  Is the rescue parameter sensitive?
>
> NSR introduces only one additional hyperparameter, the rescue window $\delta$. Empirically, $\delta$ is not highly sensitive within a moderate operating range **(table link https://postimg.cc/dk08cC2P)**. NSR consistently improves over the DAPO baseline for $\delta \in [0.05, 0.3]$, with the most reliable gains appearing around $\delta=0.1$ or $0.2$. When $\delta$ becomes too large (e.g., $0.4$ or $0.5$), performance starts to degrade. This is consistent with the intended role of $\delta$: it controls the width of the rescue window, so larger values rescue a wider near-boundary region, while overly large values may begin to admit more deeply OOB tokens. In practice, we found that a moderate $\delta$, chosen roughly **in proportion to the underlying clip width, works well without heavy tuning.**
>
> ---
>
> Reference:
>
> [1] DAPO: An Open-Source LLM Reinforcement Learning System at Scale

---

> > ### Author Rebuttal · Reviewer_uHFn · 2026-04-02
> >
> > Thank for the author's efforts and my concerns have been partially resolved. Since some concerns still remain, I tend to maintain the current scores.
> >
> >
> > 1.  Similar to prior works such as DAPO, this paper aims to mitigate the side effects of the clipping operator in RLVR training by introducing a smoother clipping strategy using a random function instead of hard clipping. However, recent work [1] suggests that the clipping operator may be unnecessary in RLVR when a KL divergence between $\pi_{\theta}$ and $ \pi_{old} $ is incorporated. Moreover, [2] demonstrates strong performance using objectives without any clipping. Since this paper does not compare against methods that remove the clipping operator, it remains unclear whether the proposed, more complex clipping design provides meaningful benefits.
> >
> > 2. In line 739 under “General Optimization Settings,” the global batch size is set to 512 with a minibatch size of 32, which is an uncommon experimental configuration. Such a setting amplifies the side effects of the hard clipping operator.
> >
> >
> > [1] https://arxiv.org/abs/2505.12366
> >
> > [2] https://arxiv.org/abs/2504.02546

---

> > > ### Author Response · Authors · 2026-04-03
> > >
> > > Thank you for your continued engagement and  follow-up questions. We fully understand your perspective and welcome this opportunity to further clarify the scope of our research and the rationale behind our experimental settings.
> > >
> > >
> > > ---
> > >
> > > ### Q1: Is the clipping operation still necessary after introducing KL divergence, and what is the significance of our method?
> > >
> > > We agree that stabilizing RLVR through KL regularization, or even removing clipping altogether, is an important research direction, and we appreciate the reviewer for pointing us to these works.
> > >
> > > That said, we do not believe this invalidates our problem setting. More broadly, the field has evolved along **TRPO → PPO → GRPO**, and recent RLVR methods appear to branch into **two parallel directions**:
> > > - One direction **retains clipping** and improves stability through trust-region boundary design, ratio shaping, sampling strategy, and boundary softening;
> > > - The other direction **weakens or removes clipping** and instead controls policy drift via KL regularization or other unclipped objectives.
> > >
> > > Our paper clearly belongs to the former line. Specifically, we study whether, within clipping-based RLVR objectives, the rigid binary decision of hard clipping systematically discards useful near-boundary signals.
> > >
> > > From this perspective, the KL-only / no-clipping methods raised by the reviewer are better viewed as **parallel and potentially complementary** directions, rather than counterexamples that directly refute our setting. In fact, improving clipping / boundary behavior is not a niche corner case, but an **active and systematized line of RLVR research**. For exactly this reason, our related-work section already places the paper within this line and discusses DAPO[2], GSPO[3], CISPO[4], as well as SAPO[5], ASPO[6], BAPO[7] and PSPO[8], all of which improve RLVR from the perspective of clipping or boundary behavior.
> > >
> > > Therefore, we believe the **more appropriate conclusion is not ‘once no-clipping objectives exist, clipping no longer deserves study,’ but rather that RLVR currently contains two parallel lines of development**, and this paper focuses on the widely used clipping-based family.”
> > >
> > > We will revise the paper to make the scope more explicit, while KL-based / no-clipping methods will be discussed as an important parallel line of work. Moreover, in our rebuttal experiments, NSR still **brings additional gains** on top of Clip-Cov / KL-Cov, further suggesting complementarity rather than conflict.
> > >
> > > ---
> > >
> > > ### Q2: Is the setting of Global batch size 512 and Minibatch size 32 "uncommon"?
> > >
> > > We deeply respect your intuition, but we must explicitly clarify that for RLVR training, a Global Batch Size of 512 combined with a Minibatch Size of 32 is by no means an uncommon configuration. On the contrary, it is the industry-recognized, de facto standard recommended by official frameworks, consistent with **the most mainstream open-source solutions** today (e.g., DeepSeek-R1[1] and DAPO[3] and VeRL framwork[9]).
> > >
> > > - First, at the micro-implementation level: This is the official default configuration of VERL, one of the most authoritative open-source RLVR frameworks today. **In VERL's officially released DAPO/GRPO training scripts**, the 512/32 parameter setting is strictly utilized (see official repository script: https://github.com/verl-project/verl-recipe/blob/ba246418f4de12b845a09bba975f1a5242adc898/dapo/test_dapo_7b.sh).
> > > - Second, at the macro-scaling level: This ratio perfectly aligns geometrically with the training hyperparameters of current frontier models. The official **DeepSeek-R1 technical report [1]** explicitly states the use of a Global Rollout Batch Size of 8192 and a training Minibatch Size of 512. The ratio between the two is strictly 16 ($8192 / 512 = 16$). The experiments we conducted are scaled exactly according to this proportion, strictly preserving the identical multiplier ($512 / 32 = 16$).
> > >
> > >
> > > Therefore, the side effects of clipping we observed do not stem from extreme or unreasonable hyperparameter choices, but are genuinely present within **the standard training paradigms** widely adopted by the community today.
> > >
> > > ---
> > >
> > >
> > > We hope this clarifies our methodological choices and reassures you of the soundness of our experimental baseline. We deeply appreciate your time and effort in helping us improve this paper.
> > >
> > > [1] https://arxiv.org/abs/2501.12948
> > >
> > > [2] https://arxiv.org/abs/2503.14476
> > >
> > > [3] https://arxiv.org/abs/2507.18071
> > >
> > > [4] https://arxiv.org/abs/2506.13585
> > >
> > > [5] https://arxiv.org/abs/2511.20347
> > >
> > > [6] https://arxiv.org/abs/2510.06062
> > >
> > > [7] https://arxiv.org/abs/2510.18927
> > >
> > > [8] https://arxiv.org/abs/2411.11681
> > >
> > > [9] https://github.com/verl-project/verl

---

### Official Review · Reviewer_2Ut7 · 2026-03-12

**Soundness:** 2
**Presentation:** 2
**Significance:** 2
**Originality:** 2
**Overall Recommendation:** 4
**Confidence:** 3

**Summary:**

This paper studies optimization instability in GRPO-style RLVR and argues that the main bottleneck is the hard clipping decision itself: tokens just outside the clipping boundary are dropped entirely, even when they may still contain useful learning signal. To address this, the paper proposes Near-boundary Stochastic Rescue (NSR), a plug-in modification that stochastically rescues slightly out-of-bound updates by perturbing the execution ratio while keeping the original clipping judgment. The paper provides an expectation-level interpretation showing that NSR behaves like an implicit soft clipping rule with $\frac{1}{r^2}$ gradient attenuation in the rescue zone, and reports gains on several benchmarks.

**Compliance With Llm Reviewing Policy:**

Affirmed.

**Final Justification:**

The authors added experiments solve some of my main concerns so I increased my score to 4

**Key Questions For Authors:**

See weakness.

**Limitations:**

Yes

**Strengths And Weaknesses:**

**Strength:** The paper addresses a real problem. In PPO/GRPO-style RL for reasoning, clipping does discard gradients abruptly, and the question of whether useful signal exists near the boundary is important. The paper also includes a diagnosis section, a targeted rescue-vs-push-out ablation, and a stability comparison against deterministic decay, which together make the paper more thoughtful. The empirical gains over DAPO and GSPO are often nontrivial, especially on the smaller models.



**Weakness:**  The paper’s central diagnosis is not as well established as it claims. The argument that “the bottleneck is the decision, not the magnitude” mainly relies on synthetic interventions, but these do not rule out other sources of instability or justify the broader claim that hard clipping is the main bottleneck in GRPO-style RLVR. The “clean decision” experiment also already introduces a stochastic rescue mechanism very similar to NSR, which weakens the claimed separation between diagnosis and method.  A stronger diagnostic would be to directly plot each token’s distance from the clipping boundary under the clean ratio, and the actual value of recovering that token, such as its positive update contribution, downstream reward improvement, or a replay-based counterfactual gain. If the paper’s diagnosis is correct, this analysis should reveal a clear pattern: deeply out-of-bound tokens should have little value, while slightly out-of-bound tokens should remain highly valuable, with the recoverable value decaying as the token moves farther away from the boundary.

The paper appears to keep the clipping thresholds fixed rather than sweeping over different clip ratios. As a result, the paper supports only the narrower claim that, under these particular clipping settings, some near-boundary out-of-bound tokens still contain useful learning signal. It does not show that boundary-adjacent tokens are valuable for a broad range of clip ratios, nor that this is a universal phenomenon across different trust-region widths. A natural missing ablation would therefore be a clip-ratio sweep to test whether the value of rescued tokens systematically changes as the clipping boundary becomes tighter or looser.

The theory is descriptive rather than conclusive: it characterizes the proposed operator, but does not show convergence, a better bias–variance tradeoff, or why this approach should outperform other smooth trust-region alternatives.

The math benchmarks are limited to small testsets, it is better to incoporate larger test sets such as MATH and
minervamath. Only Qwen model class is tested.

---

> ### Author Rebuttal · Authors · 2026-03-30
>
> We sincerely appreciate your constructive feedback. Below we address each concern in detail.
>
> > On the diagnosis and the value of out-of-bound (OOB) tokens
>
> We sincerely thank the reviewer for the constructive critique. We will revise the manuscript accordingly to clarify that hard clipping should be viewed as **an important and practically consequential factor**, rather than the unique bottleneck.
>
> **1. Clarifying the diagnosis and the role of the “clean decision” probe.**
> Our goal is not to claim that the binary clipping decision is the sole source of instability, but to isolate its effect from that of update magnitude. In this sense, the “clean decision” experiment in Sec. 3.3 is a synthetic diagnostic probe, rather than the proposed training rule itself. NSR is the practical method motivated by this diagnosis: a conservative, boundary-local rescue mechanism that only targets slightly OOB tokens.
>
> **2. Direct evidence that recoverable value is concentrated near the boundary.**
> We agree that a stronger test is to examine how the utility of recovering OOB tokens varies with boundary distance. To address this, we add two new analyses.
>
> First, we plot the **discarded signal against distance** to the clipping boundary and find that the discarded mass is concentrated near the boundary on both the positive- and negative-advantage sides **(fig link https://postimg.cc/kDwkyvH5)**.
>
> Second, we conduct a **replay-style counterfactual continuation study** with controlled OOB recovery windows. Since the effect of recovering a token is often **too small to distinguish in a single step**, we measure its accumulated downstream impact over multiple training steps **(fig link https://postimg.cc/N9DPpQpb)**. The results show a clear monotonic trend: rescuing slightly OOB tokens yields the largest gain, while the benefit decreases with boundary distance, and sufficiently far-OOB tokens provide little  benefit .
>
> ---
>
> > W2: On the missing clip-ratio sweep
>
> In the paper, we fixed the clipping thresholds to the **best-performing tuned settings** used by DAPO and GSPO (see Appendix B.3) [1,2]. This is intentional: evaluating NSR on top of each baseline’s strongest setting allows us to **attribute the gain to NSR itself**, rather than to correcting a poorly chosen clip ratio.
>
>
> If improvements were shown only under suboptimal clipping thresholds, it would remain unclear whether NSR addresses a structural limitation of hard clipping or merely compensates for baseline mis-tuning. To address the reviewer’s concern, we now add a clip-ratio sweep around the baseline setting **(table link https://postimg.cc/N9F1vYQR)**. The new results show that NSR consistently improves over the matched baseline across the tested clipping range.
>
> ---
>
> > W3: On theory and why stochastic rescue can outperform deterministic decay
>
> Our core point is simple: In large autoregressive optimization, **a fixed attenuation rule can become a stable pattern that the model repeatedly adapts to and relies on**. NSR helps not just because it down-weights OOB updates, but because it makes boundary rescue **stochastic in a dropout-like manner** rather than deterministic.  Deterministic decay therefore keeps slightly OOB directions persistently available through small but reliable weights. NSR breaks this persistence: rescue is distance-aware and stochastic, so useful near-boundary signal is preserved only in expectation, while far-OOB directions remain strongly suppressed.
>
> This mechanism matters because the OOB region is not uniformly useless. Our ablations show that near-boundary OOB tokens still contain **valuable signal** (e.g., Only-Rescue in Sec. 3.3 and Binary Admission in Sec. 6.2), and that this value is **distance-dependent**: tokens closer to the boundary are more likely to be worth rescuing, while farther OOB tokens are more likely to be unreliable. NSR is designed exactly around this structure, with rescue probability decaying with boundary distance rather than indiscriminately relaxing the trust region. Our current theory characterizes this operator rather than proving formal superiority, but this mechanism is **consistent with the empirical results** under the tested setup.
>
> ---
>
> > W4: More benchmark
>
> We appreciate this suggestion. During the rebuttal period, we expanded the evaluation to include MATH500, MinervaMath and OlympicBench, and the added results continue to show consistent gains for NSR over the corresponding baselines **(table link https://postimg.cc/4n1fttGq)**. Due to the limited rebuttal timeline, we prioritized expanding dataset coverage first.
>
> ---
>
> Reference:
>
> [1] DAPO: An Open-Source LLM Reinforcement Learning System at Scale
>
> [2] Group Sequence Policy Optimization

---

> > ### Author Rebuttal · Reviewer_2Ut7 · 2026-04-05
> >
> > Thanks for the detailed rebuttal. I appreciate that the authors added more direct evidence, especially the boundary-distance analysis, the continuation study, the clip-threshold comparison, and the expanded benchmarks. These additions make the mechanism more plausible.
> >
> > However, I still view the evidence as only partial rather than conclusive. Firstly, the discarded-signal plot shows concentration of mass, but mass is not the same as utility. Second, the clip-ratio sweep is helpful but limited: it adds only one additional clipping configuration, so the robustness claim across trust-region widths remains only partially supported. Third, the paper and rebuttal does not directly compare against what seems like the most natural alternative baseline, namely a small boundary-widening. Given the assumption that boundary tokens are the most helpful, what if we just expand the clip boundary a bit? Does this also improve the performance? I understand this is different from the stochastic rescue proposed in this paper but it could be a natural baseline to strength the paper to show the proposed method is more beneficial. Finally, the new benchmark results are appreciated, but they remain within the Qwen family. On balance, my concern is reduced, and I will increase my score to 3.

---

> > > ### Author Response · Authors · 2026-04-06
> > >
> > > It is our great fortune to encounter a reviewer who really understands RLVR. In this response, we want to speak with candor about the most authentic considerations behind our experiments.
> > >
> > > Before addressing your specific questions, we want to first clarify the **core bottom line** for all our experiments: the nature of RLVR training is extremely fragile and highly prone to collapse. We have always firmly believed that **any meaningful comparison must be built upon a baseline that can be "healthily reproduced and functions normally."** Winning numerically against a baseline that has already suffered from entropy collapse or severe oscillation is completely **meaningless (do not work)**. This is exactly why we omitted certain experiments in our previous rebuttal.
> > >
> > > To answer your precise follow-up questions, here are our candid responses:
> > >
> > > ---
> > >
> > >
> > > > 1. Regarding the distinction between the "Mass" and "Utility" of discarded signals
> > >
> > > This is precisely why we insisted on combining the "histogram (which corresponds to the 'mass' you mentioned)" with the "replay-style counterfactual experiment (to prove actual utility)" in our first rebuttal. We invite you to **take another look at the counterfactual experiment (fig link: https://postimg.cc/N9DPpQpb) which might have been overlooked**. It directly quantifies the actual downstream utility and shows a clear monotonic trend: only near-boundary OOB tokens bring significant performance gains, and this gain decays rapidly as the distance from the boundary increases.
> > >
> > > ---
> > >
> > > > 2. Regarding the sweep range of clipping thresholds
> > >
> > > To be completely frank, we did test **different threshold** parameters in practice. However, the reality is that under those unreported parameter settings, the DAPO baseline simply could not work (exhibiting irreversible entropy collapse or explosion). We chose not to report these data because we firmly believe **comparisons must be anchored on a baseline that actually works** (please see the entropy collapse or explosion link: https://postimg.cc/JHn62Y1F).
> > >
> > > ---
> > >
> > > > 3. Regarding the "Boundary-widening" baseline
> > >
> > > This is an absolutely crucial question. In fact, when NSR first yielded positive results, our team's immediate internal reaction was exactly **the same as yours: "Are we just covertly widening the boundary?"**
> > >
> > > To figure this out, we specifically ran a control experiment **(fig link https://postimg.cc/06kNF7wV)**. We forcibly widened the hard clipping boundary to [0.79, 1.296], a parameter setting we found to produce **an effective clipping rate perfectly identical to that of NSR**. This ensured the only variable was the clipping mechanism itself (deterministic widening vs. stochastic rescue). The result was crystal clear: even with this perfectly matched widened boundary, the final performance and training stability were still far inferior to our NSR.
> > >
> > > ---
> > >
> > > > 4. Regarding evaluation solely on the Qwen series
> > >
> > > We also want to be honest with you about our current objective dilemma. We did indeed try other mainstream open-source models **(but RLVR simply didn't work on them)**. The objective reality right now is that the vast majority of **open-source RLVR works in the community that can run stably rely heavily on the Qwen architecture**[1-5]. This is not us deliberately cherry-picking models, but rather a universal challenge currently facing the open-source RLVR field.
> > >
> > > ---
> > >
> > > Thank you again for your professional perspective. Every point you raised **resonated with us**. However, we also felt it was necessary to candidly explain **the scientific bottom line we insist on upholding**. We apologize that due to time constraints, the figures provided this time are not fully polished.
> > >
> > > Reference:
> > >
> > > [1] https://arxiv.org/abs/2503.24290
> > >
> > > [2] https://arxiv.org/abs/2506.10947?
> > >
> > > [3] https://arxiv.org/abs/2603.22446
> > >
> > > [4] https://arxiv.org/abs/2506.14245
> > >
> > > [5] https://github.com/verl-project/verl

---

### Official Review · Reviewer_jwuH · 2026-03-12

**Soundness:** 3
**Presentation:** 3
**Significance:** 3
**Originality:** 3
**Overall Recommendation:** 4
**Confidence:** 3

**Summary:**

This paper studies instability in GRPO/DAPO-style RLVR training and argues that the main issue is not gradient magnitude itself, but the rigid binary clipping decision that discards potentially useful near-boundary signals. Based on this diagnosis, the paper proposes Near-boundary Stochastic Rescue (NSR), a simple modification that stochastically recovers slightly out-of-bound tokens while leaving in-bound and clearly out-of-bound cases largely unchanged. The paper combines controlled interventions, targeted ablations, an expectation-level derivation showing an implicit soft-clipping effect, and experiments across 7B–30B dense/MoE models with DAPO and GSPO. Empirically, NSR improves training stability and usually improves benchmark performance on math reasoning and several general reasoning evaluations.

**Compliance With Llm Reviewing Policy:**

Affirmed.

**Key Questions For Authors:**

Please note the weakness part that I listed above.

**Limitations:**

Yes

**Strengths And Weaknesses:**

#### Strengths
- Clear problem framing: the paper targets an important and practical instability issue in RLVR training.
- Strong diagnosis: the judge/executor decomposition and rescue-vs-push-out intervention make the proposed explanation more convincing than a standard heuristic paper.
- Simple method design: NSR is lightweight, local to the clipping boundary, and easy to add to existing DAPO/GSPO pipelines.
- Solid empirical coverage: experiments include multiple scales (7B/8B/30B), dense and MoE backbones, and both DAPO and GSPO.
- Useful training analysis: repeated-run results and training-dynamics plots suggest gains come from better optimization stability, not only benchmark variance.

#### Weaknesses
- The paper does not directly compare against recent methods that also soften or adapt clipping near the boundary, so the practical advantage of NSR over the nearest alternatives is still unclear.
- Current evidence supports that hard clipping is an important source of instability in the studied setup, but does not fully establish it as the primary bottleneck for GRPO-style RLVR in general.
- Some 30B results regress relative to baseline, but the paper does not explain what model scale, reward regime, or optimization condition makes NSR less effective.
- The derivation mainly provides an expectation-level soft-clipping interpretation, but does not analyze convergence, gradient variance, or why stochastic rescue should be preferable to deterministic smoothing.
- The main paper does not adequately characterize sensitivity to the rescue window $\delta$, clip thresholds, or other training hyperparameters, which weakens the plug-and-play claim.

---

> ### Author Rebuttal · Authors · 2026-03-30
>
> We sincerely appreciate your constructive feedback. Below we address each concern in detail.
>
> ---
>
> > W1: Comparison with recent soft clipping alternatives
>
> We agree that a direct comparison with recent boundary-softening alternatives would make the practical position of NSR clearer. In the submission, we already compared NSR against a generalized family of deterministic soft-decay / soft-clipping variants in Sec. 6.2, where the stochastic formulation showed stronger performance.
>
> | Method | AIME24 | AIME25 |
> | :--- | :---: | :---: |
> | DAPO | 35.83 | 15.90 |
> | SAPO | 34.30 | 16.02 |
> | NSR | **40.76** | **17.57** |
>
> To further address this point, we additionally implemented SAPO[1] during the rebuttal period using the hyperparameter setting recommended in the SAPO paper. Under the matched training setup, NSR performs better than SAPO:
>
>
> ---
>
> > W2: Scope of the “primary bottleneck” claim
>
> We agree that the current wording is stronger than what our evidence fully supports, and we will narrow this claim in the revision. A more precise statement is: **in the GRPO-style clipping-based RLVR setups studied in this paper, the rigid binary clipping decision is an important and practically consequential source of instability**.
>
> We will revise the Introduction and Conclusion accordingly, so that the paper no longer suggests that hard clipping has been established as the primary bottleneck for GRPO-style RLVR in general.
>
> ---
>
> > W3: Why some 30B results regress relative to baseline
>
> This is not a general 30B issue; the regression is localized to the **30B MoE + DAPO** setting. On the same 30B MoE backbone, NSR remains effective under GSPO, while the mixed behavior appears specifically under **token-level DAPO without router-replay-style stabilization**.
>
> Our interpretation is that this comes from the interaction between **MoE routing** and **token-level clipping** [2,3]. In MoE training, router assignments can shift during optimization, which makes token-wise ratios more brittle under token-level clipping. By contrast, **GSPO uses sequence-level clipping**, and on the same 30B MoE model it shows a much more stable pattern, which is consistent with greater robustness to token-level routing fluctuations.
>
> ---
>
> > W4: Why stochastic rescue can outperform deterministic smoothing
>
> Our core point is simple: In large autoregressive optimization, **a fixed attenuation rule can become a stable pattern that the model repeatedly adapts to and relies on**. NSR helps not just because it down-weights OOB updates, but because it makes boundary rescue **stochastic in a dropout-like manner** rather than deterministic.  Deterministic decay therefore keeps slightly OOB directions persistently available through small but reliable weights. NSR breaks this persistence: rescue is distance-aware and stochastic, so useful near-boundary signal is preserved only in expectation, while far-OOB directions remain strongly suppressed.
>
> This mechanism matters because the OOB region is not uniformly useless. Our ablations show that near-boundary OOB tokens still contain **valuable signal** (e.g., Only-Rescue in Sec. 3.3 and Binary Admission in Sec. 6.2), and that this value is **distance-dependent**: tokens closer to the boundary are more likely to be worth rescuing, while farther OOB tokens are more likely to be unreliable **(replay-style counterfactual continuation study link https://postimg.cc/N9DPpQpb)**. NSR is designed exactly around this structure, with rescue probability decaying with boundary distance rather than indiscriminately relaxing the trust region. Our current theory characterizes this operator rather than proving formal superiority, but this mechanism is **consistent with the empirical results** under the tested setup.
>
> ---
>
> > W5: Sensitivity of rescue window $\delta$
>
> NSR is designed to be lightweight in both implementation and tuning and introduces only one additional hyperparameter, the rescue window $\delta$. Empirically, $\delta$ is not highly sensitive within a **moderate operating range (table link https://postimg.cc/dk08cC2P)**. NSR consistently improves over the DAPO baseline for $\delta \in [0.05, 0.3]$, with the most reliable gains appearing around $\delta=0.1$ or $0.2$. When $\delta$ becomes too large (e.g., $0.4$ or $0.5$), performance starts to degrade. This is consistent with the intended role of $\delta$: it controls the width of the rescue window, so larger values rescue a wider near-boundary region, while overly large values may begin to admit more deeply OOB tokens. In practice, we found that a moderate $\delta$, chosen roughly **in proportion to the underlying clip width, works well without heavy tuning.**
>
> ---
>
> Reference:
>
> [1] Soft Adaptive Policy Optimization
>
> [2] Group Sequence Policy Optimization
>
> [3] Stabilizing MoE Reinforcement Learning by Aligning Training and Inference Routers

---

> > ### Author Rebuttal · Reviewer_jwuH · 2026-04-03
> >
> > My questions have been answered. I choose to keep my score.

---

> > > ### Author Response · Authors · 2026-04-03
> > >
> > > We sincerely appreciate your continued support and the positive score.
> > >
> > > If you have any further questions as indicated by the system flag, or if any new thoughts come up, please do not hesitate to share them. We remain fully available to address any additional points.

---

### Official Review · Reviewer_mE6U · 2026-03-13

**Soundness:** 3
**Presentation:** 3
**Significance:** 3
**Originality:** 3
**Overall Recommendation:** 4
**Confidence:** 3

**Summary:**

This paper identifies a critical bottleneck in Reinforcement Learning with Verifiable Rewards (RLVR) optimization, particularly in GRPO-style objectives like DAPO. The authors argue that the standard hard-clipping mechanism, designed to constrain policy updates within a trust region, acts as a rigid binary gate. This indiscriminately discards tokens whose importance ratios fall just outside the clipping boundary, even if they carry high-value learning signals. Through systematic causal interventions, the paper demonstrates that training instability stems from this clipping decision rather than from the magnitude of gradients. To address this, the authors propose Near-boundary Stochastic Rescue (NSR), a minimal, plug-and-play modification. For tokens slightly outside the trust region, NSR stochastically rescues them by applying a bounded multiplicative noise to their importance ratio, allowing their gradients to flow back into the optimization if the perturbed ratio falls within bounds. Theoretically, NSR induces an implicit soft-clipping effect with an approximate $1/r^2$ gradient decay in expectation. Extensive experiments across model scales (7B to 30B) and architectures (dense and MoE) on math and general reasoning benchmarks show that NSR consistently improves training stability and final performance over strong baselines like DAPO and GSPO.

**Compliance With Llm Reviewing Policy:**

Affirmed.

**Key Questions For Authors:**

1. Stochasticity vs. Deterministic Softmax: The results in Table 3 show that deterministic "Explicit Decay" helps but is unstable. Can you provide more intuition or analysis on why the stochastic admission in NSR is particularly effective at suppressing "transient noise" and preventing "over-commitment to outliers"? Is it akin to a form of dropout or noise injection specifically at the decision boundary?
2. Rescue Window ($\delta$) Tuning: How should practitioners set the $\delta$ parameter for a new task or algorithm beyond the provided defaults? Is it loosely tied to the clipping threshold $\epsilon$? Did you observe any performance cliffs or sharp transitions when varying $\delta$?
3. Generalization to Other Clipping Paradigms: The paper shows NSR works for both token-level (DAPO) and sequence-level (GSPO) clipping. Do you foresee any challenges or necessary adaptations for applying NSR to other policy constraint mechanisms, such as those using explicit KL penalties?

**Limitations:**

1. The method and analysis are specific to the RLVR paradigm with verifiable (e.g., binary) rewards and group-based advantage estimation. Its effectiveness in standard RL with learned value functions and continuous advantage estimates is not validated.
2. The computational budget for experiments is substantial (hundreds of thousands of GPU hours), which, while a strength for credibility, also means the approach's cost-effectiveness for smaller-scale research is implied but not explicitly compared.
3. The "minimal fix" claim is slightly nuanced by the introduction of a new hyperparameter ($\delta$), however small its tuning space might be.

**Strengths And Weaknesses:**

Strengths:
- Clear and Convincing Diagnosis: The core strength lies in the rigorous, causal diagnosis of the problem. The decoupling of the "decision" and "execution" roles of the importance ratio (Sec. 3.2) and the subsequent targeted ablations (Sec. 3.3, 4.2) provide compelling evidence that the hard-clipping decision is the primary bottleneck. This moves beyond common assumptions about gradient magnitude.
- Elegant and Simple Solution: NSR is a minimal, intuitive intervention directly motivated by the diagnosis. Its plug-and-play nature, with negligible overhead, is a significant practical advantage.
- Thorough Empirical Validation: The evaluation is extensive, covering multiple model families, scales, architectures, and benchmark types (math and general knowledge). The consistent gains demonstrate robustness and generalizability. The analysis of training dynamics (stability, entropy, length) is valuable.
- Strong Ablations and Mechanistic Analysis: The paper goes beyond just showing NSR works. The layered ablation in Sec. 6.2 effectively disentangles the contributions of signal admission, deterministic decay, and stochasticity, providing deeper insight into why NSR is effective.

Weaknesses:
- Theoretical-Empirical Gap: While the theoretical derivation of the expected gradient decay ($O(1/r^2)$) is sound, the ablation studies clearly show that deterministic soft decay with a similar profile is insufficient to match NSR's performance. The paper attributes the superior performance to the "stochastic robustness" of NSR, but this mechanism could be explored more deeply. Why is stochastic filtering near the boundary more robust than deterministic attenuation?
- Limited Exploration of Hyperparameter Sensitivity: The success of NSR depends on the rescue window $\delta$. While default values are provided for DAPO and GSPO, a more systematic analysis of the sensitivity to $\delta$ across different tasks and model sizes would strengthen the practical guidance.
- Potential Downsides of Stochasticity: Introducing stochasticity, even bounded, adds variance to the gradient estimate. The paper shows improved stability, but a discussion on the potential trade-offs or scenarios where this variance might be detrimental would be balanced.

---

> ### Author Rebuttal · Authors · 2026-03-30
>
> We sincerely appreciate your constructive feedback. Below we address each concern in detail.
>
> ---
>
> > W1 & Q1: Why stochastic rescue is more robust than deterministic attenuation
>
>  Our core point is simple: In large autoregressive optimization, **a fixed attenuation rule can become a stable pattern that the model repeatedly adapts to and relies on**. NSR helps not just because it down-weights OOB updates, but because it makes boundary rescue **stochastic in a dropout-like manner** rather than deterministic.  Deterministic decay therefore keeps slightly OOB directions persistently available through small but reliable weights. NSR breaks this persistence: rescue is distance-aware and stochastic, so useful near-boundary signal is preserved only in expectation, while far-OOB directions remain strongly suppressed.
>
> This mechanism matters because the OOB region is not uniformly useless. Our ablations show that near-boundary OOB tokens still contain **valuable signal** (e.g., Only-Rescue in Sec. 3.3 and Binary Admission in Sec. 6.2), and that this value is **distance-dependent**: tokens closer to the boundary are more likely to be worth rescuing, while farther OOB tokens are more likely to be unreliable **(replay-style counterfactual continuation study link https://postimg.cc/N9DPpQpb)**. NSR is designed exactly around this structure, with rescue probability decaying with boundary distance rather than indiscriminately relaxing the trust region.
>
> ---
>
> > W2, Q2 & Limitation 3: Rescue window tuning
>
> NSR introduces only one additional hyperparameter, the rescue window $\delta$. Empirically, $\delta$ is not highly sensitive within a **moderate operating range (table link https://postimg.cc/dk08cC2P)**. NSR consistently improves over the DAPO baseline for $\delta \in [0.05, 0.3]$, with the most reliable gains appearing around $\delta=0.1$ or $0.2$. When $\delta$ becomes too large (e.g., $0.4$ or $0.5$), performance starts to degrade. This is consistent with the intended role of $\delta$: it controls the width of the rescue window, so larger values rescue a wider near-boundary region, while overly large values may begin to admit more deeply OOB tokens. In practice, we found that a moderate $\delta$, chosen roughly **in proportion to the underlying clip width, works well without heavy tuning.**
>
> ---
>
> > W3: Potential downsides of stochasticity
>
> We agree that stochasticity can in principle increase gradient variance. However, in our setting we do not observe evidence that bounded noise is itself detrimental to training. To isolate this factor, we repeated the **DAPO + Advantage Scalar Noise intervention**(Sec. 3.1) three times **(fig link https://postimg.cc/0Kq49xCL)**. This intervention injects multiplicative noise into the update magnitude while keeping the clipping decision unchanged, and the resulting runs show very similar performance and training dynamics to the DAPO baseline. This suggests that **bounded stochasticity by itself does not destabilize optimization** in the studied setup.
>
> NSR is even **more conservative** than this global noise intervention. Its perturbation is **bounded by the rescue window** $\delta$ and applied only to a small subset of near-boundary OOB tokens, while most in-bound updates remain unchanged. Thus, NSR introduces only localized stochasticity around the **clipping boundary rather than global optimization noise**. We will clarify this trade-off in the revision: although stochasticity can in principle increase variance, our experiments show no harmful effect from bounded, boundary-local noise, and the repeated-run results further support the practical stability of NSR.
>
> ---
>
> > Q3 & Limitation 1：generalization scope
>
> Our claims are intentionally **scoped to the RLVR setting** studied in this paper. The diagnosis, analysis, and NSR mechanism are all developed for clipping-based RLVR. We do not claim that the same instability source, or the same fix, automatically transfers to standard RL settings with learned value functions and continuous advantage estimates.
>
> ---
>
> > Limitation 2: Computational budget vs. method overhead
>
> We believe it is important to distinguish the **incremental cost of NSR itself** from the **total compute used for experimental validation**. NSR is a minimal plug-and-play modification: it only applies a bounded random perturbation to a small subset of near-boundary tokens, so its additional per-step training overhead is negligible relative to standard DAPO/GSPO training.
>
> The substantial GPU budget in the paper mainly reflects the cost of rigorous evaluation, not the cost of NSR itself. In the early stage of this work, we repeated some settings many times to verify the consistency of the observed effect, and the main reported results are averaged over multiple full training runs to ensure reliability. We therefore view the high total compute as a consequence of careful experimental validation rather than as a limitation specific to the NSR algorithm.

---

> > ### Author Rebuttal · Reviewer_mE6U · 2026-04-04
> >
> > None

---

> > > ### Author Response · Authors · 2026-04-04
> > >
> > > We are glad that our rebuttal fully addressed your questions. We appreciate your positive score and thoughtful review.

---

### Decision · Program_Chairs · 2026-04-30

**Decision:**

Accept (regular)

**Comment:**

All four reviewers (mE6U: 4, jwuH: 4, 2Ut7: 4, uHFn: 4) converged on weak accept, representing a clear consensus. The paper proposes Near-boundary Stochastic Rescue (NSR), a lightweight plug-and-play modification to hard-clipping in GRPO-style RLVR that stochastically retains near-boundary out-of-bound tokens. Reviewers mE6U and jwuH both praised the "clear and convincing diagnosis" separating the clipping decision from gradient execution, as well as the elegance and simplicity of the proposed fix. Reviewer 2Ut7 raised legitimate concerns about the strength of the diagnostic evidence and benchmark coverage, which the authors addressed through a replay-style counterfactual continuation study and expanded evaluations on MATH500, MinervaMath, and OlympicBench; 2Ut7 upgraded from 2 to 4 as a result. Reviewer uHFn's remaining concern is addressed reasonably by the authors: clipping-based and KL-based RLVR represent parallel active directions, and NSR demonstrably adds value on top of Clip-Cov/KL-Cov in rebuttal experiments. The authors also submitted a confidential AC comment disputing uHFn's assessment on several grounds.